# *In vitro* characterization and *in vivo* comparison of the pulmonary outcomes of *Poractant alfa* and *Calsurf* in ventilated preterm rabbits

Xiaojing Guo[1], Siwei Luo[1], Davide Amidani[2], Claudio Rivetti[3], Giuseppe Pieraccini[4], Barbara Pioselli[2], Silvia Catinella[2], Xabi Murgia[5], Fabrizio Salomone[2], Yaling Xu[1], Ying Dong[1], Bo Sun[1]*

1 Departments of Pediatrics and Neonatology, Children's Hospital of Fudan University, Shanghai, China, 2 Department of Research and Development, Chiesi Farmaceutici, Parma, Italy, 3 Department of Chemistry, Life Sciences and Environmental Sustainability, University of Parma, Parma, Italy, 4 CISM Mass Spectrometry Centre, Department of Health Sciences, University of Florence, Firenze, Italy, 5 Department of Drug Delivery, Helmholtz Institute for Pharmaceutical Research, Saarbrücken, Saarland, Germany

* bsun98@aliyun.com, bsun9898@163.com

**Data Availability Statement:** All relevant data are within the paper and its Supporting Information files.

## Abstract

*Poractant alfa* and *Calsurf* are two natural surfactants widely used in China for the treatment of neonatal respiratory distress syndrome, which are extracted from porcine and calf lungs, respectively. The purpose of this experimental study was to compare their *in vitro* characteristics and *in vivo* effects in the improvement of pulmonary function and protection of lung injury. The biophysical properties, ultrastructure, and lipid composition of both surfactant preparations were respectively analysed *in vitro* by means of Langmuir-Blodgett trough (LBT), atomic force microscopy (AFM), and liquid-chromatography mass-spectrometry (LC-MS). Then, as core pharmacological activity, both head-to-head (100 and 200 mg/kg for both surfactants) and licensed dose comparisons (70 mg/kg *Calsurf* vs. 200 mg/kg *Poractant alfa*) between the two surfactants were conducted as prophylaxis in preterm rabbits with primary surfactant deficiency, assessing survival time and rate and dynamic compliance of the respiratory system ($C_{dyn}$). Intrapulmonary surfactant pools, morphometric volume density as alveolar expansion ($V_v$), and lung injury scores were determined *post mortem*. AFM and LC-MS analysis revealed qualitative differences in the ultrastructure as well as in the lipid composition of both preparations. *Calsurf* showed a longer plateau region of the LBT isotherm and lower film compressibility. *In vivo*, both surfactant preparations improved $C_{dyn}$ at any dose, although maximum benefits in terms of $V_v$ and intrapulmonary surfactant pools were seen with the 200 mg/kg dose in both surfactants. The group of animals treated with 200 mg/kg of *Poractant alfa* showed a prolonged survival time and rate compared to untreated but ventilated controls, and significantly ameliorated lung injury compared to *Calsurf* at any dose, including 200 mg/kg. The overall outcomes suggest the pulmonary effects to be dose dependent for both preparations. The group of animals treated with 200 mg/kg of *Poractant alfa* showed a significant reduction of mortality. Compared to *Calsurf*, *Poractant*

**Funding:** The study was supported by Chiesi Farmaceutici S.p.A. The company contributed to the study design but had no influence in the performance, analysis, and interpretation of experimental data and in writing the manuscript. YD is a recipient of research grants from the National Natural Science Foundation (No. 81501288) and Shanghai Municipal Commission of Health (Project Young Physician Investigator).

**Competing interests:** The study was supported by Chiesi Farmaceutici S.p.A, which owns the marketing rights for Poractant alfa. The company contributed to the study design but had neither influence on the performance, analysis, and interpretation of experimental data nor in writing the manuscript. BP, SC, and FS are employees of Chiesi Farmaceutici S.p.A. XM served as consultant for Chiesi in this study. YD is a recipient of research grants from the National Natural Science Foundation (No. 81501288) and Shanghai Municipal Commission of Health (Project Young Physician Investigator). This does not alter our adherence to PLOS ONE policies on sharing data and materials.

*alfa* exerted better effects if licensed doses were compared, which requires further investigation.

## Introduction

Neonatal respiratory distress syndrome (RDS) is a condition of pulmonary insufficiency affecting preterm infants. With an incidence as high as 50% in very and extremely preterm infants (born below 33 and 28 weeks of gestational age, respectively), the development of RDS is associated with a high risk of perinatal morbidity and mortality.[1,2] At low gestational age, alveolar epithelial cells are structurally and functionally immature and the ability of surfactant synthesis and secretion is therefore compromised.[3] Consequently, surfactant pools in both alveolar and parenchymal compartments are noticeably low at birth,[4] leading to high intrapulmonary pressure and alveolar collapse.[5] RDS develops shortly after birth and evolves rapidly towards a life-threatening hypoxemic condition unless vigorous rescue treatment is provided to facilitate adequate alveolar expansion and intrapulmonary perfusion,[6] of which surfactant replacement therapy is the most efficient pharmacological treatment.

Animal-derived surfactants contain several classes of phospholipids, including relatively high amounts of disaturated phospholipids, namely dipalmitoyl- phosphatidylcholine (DPPC) as one of the main active components, and the hydrophobic surfactant proteins (SP)-B and SP-C.[6,7] As an emerging country, neonatal-perinatal care in China has been advanced in the past decade. With 16–17 million annual births and a preterm birth rate of 4–5%, 100,000 cases are presumed to be affected by RDS and require surfactant therapy every year in China.[8,9] *Poractant alfa* and *Calsurf* are two animal-derived surfactant preparations widely used in China for the treatment of RDS. *Poractant alfa* is extracted from porcine minced lungs and is licensed to be administered at phospholipid doses of 100 or 200 mg/kg, with salient pharmacodynamics and pharmacokinetic effects in the treatment of RDS.[10–13] Moreover, *Poractant alfa* has undergone large randomized clinical trials comparing its efficacy to other surfactant preparations.[13–16] On the other hand, little is known about *Calsurf*, which is extracted from lung washes of calves and is indicated for the treatment of RDS at an initial phospholipid dose of 40–100 (average 70) mg/kg (manufacturer information). To our knowledge, very few clinical studies addressing the pulmonary efficacy of *Calsurf* have been published to date in international, peer-reviewed journals.[17] Neither surface properties nor clinical effects compared to porcine or other bovine surfactants were provided. Only the information that *Calsurf* has a similar formulation in terms of phospholipid concentration (roughly about 30 mg/ml) as other bovine-derived surfactants has been reviewed.[18] As a comparison between the clinical efficacy of *Poractant alfa* and *Calsurf* is not available yet, we performed a direct comparison of both surfactant preparations *in vitro* and *in vivo*. Our aim was to perform a preliminary preclinical characterization of *Calsurf*, providing insights into both biophysical and physiological properties as well as the pharmacodynamics of this surfactant preparation in comparison with *Poractant alfa*.

## Materials and methods

### Surfactant preparations

*Poractant alfa* (Curosurf®, Chiesi Framaceutici, Parma, Italy) is a natural surfactant suspension, prepared from minced porcine lungs, containing almost exclusively polar lipids, in particular, phosphatidylcholine (PC, about 70% of the total phospholipid content), and about 1%

of specific low molecular weight hydrophobic proteins SP-B and SP-C, at a phospholipid concentration of 80 mg/mL (product information). *Calsurf* (Kelisu®, Shuanghe Pharmaceuticals, Huarun Group, Beijing, China) is a natural surfactant, prepared from calf lung washes, containing approximately 80% phospholipids, at least 55% phosphatidylcholine, and about 1–2% of SP-B and SP-C at a phospholipid concentration of 35 mg/mL (product information). *Calsurf* is presented as a lyophilized product and must be re-suspended with distilled water before use.

## Liquid-chromatography mass-spectrometry for phospholipids analysis

A detailed description of the method is provided in Supporting information (S1 Method). Liquid chromatography solvents, acetic acid and ammonia solution were purchased from Sigma Aldrich Italy (Milan, Italy). The chemical standards of phospholipids and the internal standards used for quantitative purposes were obtained from Avanti Polar Lipids (Alabaster, AL, USA).

Normal phase liquid chromatography mass spectrometry (NPLC-MS) was performed on a Thermo Scientific (Bremen, Germany) system composed by a high performance liquid chromatography (HPLC) surveyor, with column oven and auto-sampler, coupled to an LTQ ion trap mass spectrometer via electrosprayinterface (ESI) (Thermo). Surfactant samples were separated on a Varian Polaris Si A, 250 x 2.1 mm, 5 μm, 200 Å (Agilent Technologies, Santa Clara, CA, USA) with a ternary gradient.

Acquisition was performed by polarity switching, recording both negative and positive ions during the entire elution program. Data were acquired in full scan mode, from 200 to 1600 m/z, excluding cholesterol that was acquired in positive ion MS/MS mode, recording the two product ions at m/z 161.1 and 243.2 deriving from the precursor ion at m/z 369.3, $[M+H-H_2O]^+$ (isolation window 2.6 m/z, normalized collision energy 20). Data were acquired and elaborated by Xcalibur software version 2.0.7. Semi-quantitative data were accessed by manual peak integration of the analytes of interest.

## Atomic force microscopy measurements

A 70 mg mass of commercial lyophilized *Calsurf* powder was re-suspended in a vial using 2 mL of sterile water for injection and gently mixed by vial rotation for 20 min at room temperature. *Poractant alfa* was analyzed by AFM at a phospholipid concentration of 80 mg/mL. Further, it was diluted 2.2 times in saline solution to a final concentration of 35 mg/mL so that the results of both preparations could be directly compared at the same phospholipid concentration. Volumes of 30 μl of *Poractant alfa* and *Calsurf* were deposited onto freshly cleaved mica substrates for 2 min. Thereafter, the surface was rinsed with MilliQ water (≥18.2 MΩ-cm) and dried with a nitrogen gas flow. AFM imaging was carried out "in air" with a Nanoscope IIIA microscope (Digital Instruments, Santa Barbara, CA, USA) equipped with the J scanner and commercial silicon cantilevers (MikroMasch, Tallinn, Estonia) operating in tapping mode. Images of 512 × 512 pixels were collected with a maximum scan size of 10 μm, at a scan rate of 1 or 2 lines per second. While imaging, height, amplitude and phase signals were recorded. A surface roughness analysis was carried out using the dedicated tool of the *Gwyddion v2.53* software. The *Rtm* parameter represents the mean peak-to-valley roughness and it was determined by the difference between the highest peak ant the lowest valley within multiple samples in the evaluation length. Roughness data were collected on three independents images (10 μm x 10 μm) from independent experiments of *Poractant alfa* and *Calsurf*. For each image, 11 local roughness observations were performed, by means of an evaluation line of 10 μm in length x 100 nm in thickness, shifted along the entire image by a step of 1 μm.

## Langmuir Blodgett measurements

Both *Poractant alfa* and *Calsurf* were subjected to lipid extraction according to the method described by Bligh and Dyer.[19] The chloroform phase containing surfactant components was diluted to a final phospholipid concentration of 1 mg/mL and used for the Langmuir Blodgett trials. Before each measurement, the ultra-pure water subphase (Milli-Q $\geq$18.2 MΩ-cm) was quickly compressed to verify the absence of possible organic contaminations. Surfactant was spread onto the subphase of a ribbon barrier trough (KSV NIMA, Finland) by means of a 50 μl glass microsyringe (Hamilton Company, Bonaduz, Switzerland). The subphase operational area was 156 cm$^2$ and the temperature was maintained constant at 25 ± 0.5°C using an external circulating water bath.

The chloroform phase containing surfactant components was spread onto the water subphase to reach a surface pressure of approximately 10 mN/m. Surface pressure is the amount by which surface tension is reduced. For instance, a surface pressure of zero mN/m indicates no reduction at all of surface tension, whereas a surface pressure value of 70mN/m in a water (surface tension of water at 37° C is 70 mN/m) indicates that surface tension has been fully counteracted by surfactant.[20] After spreading, the film was left undisturbed for 15 min to allow solvent evaporation. The film was then symmetrically compressed at a rate of 40 cm$^2$/min. A surface pressure/area (π/A) isotherm was recorded in real time with the built-in software. The value of compressed area at a surface pressure of 68 mN/m was used to compare both formulations. For each surfactant, at least three independent experiments were performed. Two-dimensional compressibility of surfactant films was obtained from the compression isotherms. Film compressibility ($C_m$) is defined as the inverse of the compression modulus and is given by:

$$C_m = -1/A^*(\delta A/\delta\pi) \tag{1}$$

where A is the surface area and π is the surface pressure. $C_m$ provides information concerning phase transitions and fluidity/elasticity of the monolayer. Large $C_m$ values are indicative of the state in which the film displays high compressibility and fluidity, whereas small $C_m$ values reflect a high packing of the phospholipid molecules. For each measurement, $C_m$ was plotted as a function of the surface pressure π.

## Standardized positive inspiratory pressure (PIP) ventilation loop experiments in preterm rabbits

Seven pregnant, date-mated New Zealand White rabbits were obtained from Shanghai Songlian Experimental Animal Center and housed until the twenty-sixth day of gestation under standard conditions according to the current procedures for animal housing and handling of the center. The does were transported to the experimental site one day prior to the experiments with full shelter, food and water *ad libitum*. The study protocol was approved by the ethics committee of the Children's Hospital of Fudan University (No. 2016240), and all efforts were made to minimize animal suffering.

One the experimental days, does were sedated with 2 mL i.m. 0.5% Diazepam (Shanghai Xudong Haipu Pharmaceutical Co. Ltd., Shanghai, China) and paralyzed with 10 mL i.m. 20% Urethane (Ethyl carbamate, BBI Life Sciences, Shanghai, China), followed by intravenous catheterization for additional 10–15 mL Urethane infusion. A maintenance dosage of Urethane was given at 1 mL/kg per hour to keep does unconsciousness until the end of the cesarean section. The does were then sacrificed by an overdose of potassium chloride.

Preterm rabbit fetuses were obtained by cesarean section after 27 days of gestation (27$^{+0h}$-27$^{+6h}$ days of gestation, term 31 days). After delivery, rabbit pups were weighed and

immediately anaesthetized with 0.1 mL i.p. 1% Lidocaine hydrochloride (Shandong Hualu Pharmaceutical Co. Ltd., Shandong, China). Pups were then tracheostomized, intratracheally intubated with a thin, short metal cannula (18G needle, 1.2 mm outer diameter, 10–12 mm in length), and connected through silica tubing to a multi-plethysmograph-ventilator system.[21] The pressure delivered to each rabbit was measured by a pressure transducer (Shanghai Yangfan Electronic Co. Ltd., Shanghai, China) and tidal volume ($V_T$) was recorded by a pneumo-tachometer (RSS100-HR, Hans Rudulph, inc. Kansas City, USA). Both were connected to an automated physiologic monitoring system (PowerLab, ADInstruments Pty Ltd, Bella Vista, Australia).

In total 60 rabbit pups were allocated to one of the six experimental groups (n = 10 per group) in natural order by litter and delivery, in sequence and consecutive manner. To compare the *in vivo* effects both in licensed doses and head-to-head way, two groups of animals received intratracheal *Poractant alfa* at doses of 200 mg/kg and 100 mg/kg (P200 and P100 groups), which were 2.50 and 1.25 ml/kg as for fluid volume, and three further groups received *Calsurf* at doses of 200 mg/kg, 100 mg/kg and 70 mg/kg (C200, C100, and C70 groups), which were 5.71, 2.86 and 2.00 ml/kg as for fluid volume. A control (Ctrl) group received only sham volume of air. The rabbit pups in six groups were paralleled submitted to mechanical ventilation (Siemens 900C ventilator, Siemens-Elema, Solna, Sweden) with 100% oxygen, for 30 min according to a fixed ventilation protocol, as previously described.[22,23] Briefly, the pups were initially ventilated for 15 min with a positive inspiratory pressure (PIP) of 25 cmH$_2$O, followed by 5 min at a PIP of 20 cmH$_2$O, another 5 min at a PIP of 15 cmH$_2$O, and a final step of 5 min in which the PIP was restored to the initial level of 25 cmH$_2$O. 0.5 mL 2% lidocaine was intracranially injected at the end of ventilation.

Since positive end-expiratory pressure (PEEP) modifies the response to surfactants in ventilated immature rabbits,[24, 25] two different experimental sessions were carried out. In the first set of experiments, PEEP was not applied. In the second session, a PEEP of 2–3 cmH$_2$O was included in the ventilation protocol to the same animals in groups to see if different responses existed between the two surfactants. $V_T$, PIP and PEEP were measured at 5 min intervals for each animal. All pups were anaesthetized throughout the study. The dynamic compliance of the respiratory system ($C_{dyn}$) was derived from:

$$C_{dyn} = V_T/(PIP-PEEP) \qquad (2)$$

where $V_T$ was in mL/kg birth weight, PIP and PEEP in cmH$_2$O.

## Prophylactic surfactant treatment in standardized $V_T$ ventilation experiment in preterm rabbits

Rabbit pups were delivered and prepared for mechanical ventilation in the same way as described in the PIP ventilation loop experiments to assess survival after surfactant administration (P200, P100, C200, C100, C70, and Ctrl groups; N = 25 pups per group from another 22 litters). For this set of experiments, the ventilator was set at 40 breaths/min, with an inspiration-to-expiration ratio (I:E) of 1:2, and a fraction of inspired oxygen of 1.0. A PIP ranging between 10–25 cmH$_2$O was applied to generate a standardized $V_T$ of 4–6 mL/kg body weight. PEEP was provided at 2–3 cmH$_2$O. PIP was titrated every 3–5 minute interval during the first 30 min of ventilation, and subsequently 10–15 minute interval afterwards. The anesthesia was the same as in the PIP loop experiments, and it was maintained for preterm rabbit pups by i. p. 0.1 mL mixed solution (0.01 mL of 2% Lidocaine, 0.03 mL 5% NaHCO$_3$ and 0.06 mL 10% Glucose) at 60–90 min intervals.

Survival time was assessed at 3 h, or at an early death. Death occurring during ventilation was determined by body skin color change and confirmed by chest wall pulsation sign of

heartbeats. The determination was made continuously during ventilation. At 180 min, euthanasia was provided by intracranial injection of 0.5 mL 2% lidocaine.

## Biochemical analysis of broncho-alveolar lavage and lung tissue samples

Lung samples from 8–10 animals per group from the prophylactic surfactant treatment experiments were preserved immediately after *exitus* by broncho-alveolar lavage (BAL) with 30 ml/kg body weight of normal saline (0.9% NaCl). BAL fluid was immediately centrifuged at 2000 rpm for 15 min at 4 °C to remove cell debris. Further, BALs underwent organic extraction with methanol/chloroform (1:2) to extract the total amount of phospholipids. Similarly, lung tissue homogenates were also extracted with methanol/chloroform (1:2). Total phospholipids (TPL), disaturated phosphatidylcholine (DSPC) and total proteins (TP) were determined by the Bartlett assay,[26] the Mason's osmium digestion and aluminum tetroxide column chromatography,[27] and the Lowry's method, respectively, as previously described.[28] For these experiments, a group of delivered but non-ventilated pups (n = 9) was included as an additional control group ($C_0$).

## Lung examination

*Post mortem*, the lungs of another 13 animals per group (other than those for biochemical analysis, both were allocated randomly) from the prophylactic surfactant treatment experiments were examined for pneumothorax followed by intrapulmonary arterial perfusion fixation with 4% formalin at 10 cmH$_2$O for 30 min. The perfusion-fixed lungs were removed *en bloc* and fixed continuously in 4% formaldehyde for 72 h. The lung blocks were embedded in paraffin, cut into thin sections (5–6 μm) and stained with hematoxylin and eosin. Alveolar expansion ($V_V$) was determined based on the image-analysis of 50 microscopic fields per lung sample at 200x magnification, using a semi-quantitative theorem of lung morphometry by point-counting method. $V_V$ reflects the magnitude of average alveolar expansion using total lung parenchyma as denominator on aerated alveolar spaces of each individual animal bilateral lung.[29] $C_V$ ($V_v$) denotes the standard deviation / mean of $V_V$ for individual lungs, reflecting the homogeneity of alveolar expansion.[30] The appendix lobe of the right lung was ligated and removed for wet-to-dry lung weight (W/D) measurement before perfusion to estimate lung tissue fluid content.[31]

Lung sections were inspected by an expert pathologist in a blind manner and assessed according to lung injury scores (LIS), which was based on four items of pathological impairment in lung tissue: small airway epithelial damage (desquamation), edema, hemorrhage, and neutrophil infiltration. Severity for each item was estimated by scores of 0 for none, 1 for mild-to-moderate (item present in < 25% of the field), 2 for moderate-to-severe (item present between 25%-50% of the field), 3 for moderate-to-severe (item present between 50%-75% of the field), and 4 for severe (item predominates in > 75% of the field).[30] The sum of all items provides an overall injury severity on an individual lung basis.

## Statistical analysis

Continuous data are presented as mean and standard deviation (SD) and subjected to analysis of variance (ANOVA) followed by post hoc comparison of between-group difference with Student-Newmann-Keuls test. Categorical data are presented as ratio or percent (%) and analyzed with Chi square test, or by non-parametric ANOVA with Kruskal-Wallis test followed by Wilcoxon-Mann-Whitney test for between-group differences. The AFM and Langmuir Blodgett datasets were subjected to ANOVA and Student's *t*-test. Survival analysis was subjected to

Log-rank (Mantel-Cox) test. A *P* value <0.05 is regarded as a statistically significant difference.

## Results

### LC-MS analysis of the lipid composition of *Poractant alfa* and *Calsurf*

The relative amount of lipid species in *Poractant alfa* and *Calsurf* are summarized in Table 1. The LC-MS analysis revealed qualitative differences between surfactant preparations. Indeed, determined amounts of PC, DPPC, phosphatidylethanolamine (PE) and phosphatidylinositol (PI) were respectively 3-, 2-, 10-, and 3-fold as high in *Poractant alfa* as *Calsurf*. Moreover, the two preparations differed in phospholipid chemical properties; in *Calsurf* 65% of the phospholipid fraction was saturated, whereas *Poractant alfa* was composed of 50% of unsaturated acyl chains phospholipid with higher molecular weight. Plasmalogens were not detected in *Calsurf*, whereas they account for 4% of the total lipid composition of *Poractant alfa*. On the contrary, the presence of cholesterol was only detected in *Calsurf*, which occupied about 3% of the total lipid.

### Atomic force microscopy measurements

At any phospholipid concentration, the ultrastructure of *Poractant alfa* was characterized by overlapping phospholipid domains, which showed complex multilamellar organizations, composed of 12–15 overlapped lipid bilayers (Fig 1). Single bilayers of *Poractant alfa* showed a height ranging between 5.4–6 nm. Conversely, the aforementioned multilamellar structures were not detected in *Calsurf*, which showed a maximum of 2–3 overlapping bilayers of a lower height, in the range of 3.5–4.5 nm.

**Table 1. Lipid analysis of *Poractant alfa* and *Calsurf*.**

| Lipid class | Calsurf (n = 4) | | Poractant alfa (n = 8) | |
|---|---|---|---|---|
| | mg/mL | %/PL | mg/mL | %/PL |
| PC** | 15.0±0.2 | 43.1 | 43.3±1.2 | 49.4 |
| DPPC | 15.9±0.2 | 45.7 | 28.2±1.1 | 32.2 |
| PG | 0.7±0.1 | 2.0 | 0.8±0.2 | 0.9 |
| BMP | 0.6±0.1 | 1.7 | 0.1±0.03 | 0.1 |
| PE | 0.5±0.1 | 1.4 | 5.9±0.9 | 6.7 |
| PI | 0.9±0.2 | 2.6 | 3.0±0.8 | 3.4 |
| SM | 0.1±0.03 | 0.3 | 2.7±0.8 | 3.1 |
| PLPE | ND | ND | 3.7±0.6 | 4.2 |
| Cholesterol | 1.1±0.2 | 3.2 | ND | ND |
| | %/PL | | %/PL | |
| Saturated PL | 66.3 | | 45.6 | |
| Mono-unsaturated PL | 27.1 | | 24.9 | |
| Poly-unsaturated PL | 6.6 | | 29.5 | |

Data are expressed as mg of each lipid class (mean±SD) and weight % with respect to 1 mL of surfactant (mean values of the indicated numbers "n" of independent observations). Notice that the lipid concentration of *Calsurf* and *Poractant alfa* is 35 mg/mL and 80 mg/mL, respectively. The phospholipid species listed in the top panel were quantified using internal phospholipid standards. Abbreviations: PC, phosphatidylcholine; DPPC, dipalmitoylphosphatidylcholine; PG, phosphatidylglycerol; BMP, bis (monoacylglycero) phosphate; PE, phosphatidylethanolamine; PI, phosphatidylinositol; PL, phospholipids; SM, sphingomyeline; PLPE, 1-palmitoy l,2-linoleoylphosphatidyl-ethanolamine

**, total content of PC with the exception of DPPC, expressed individually; ND, not detectable.

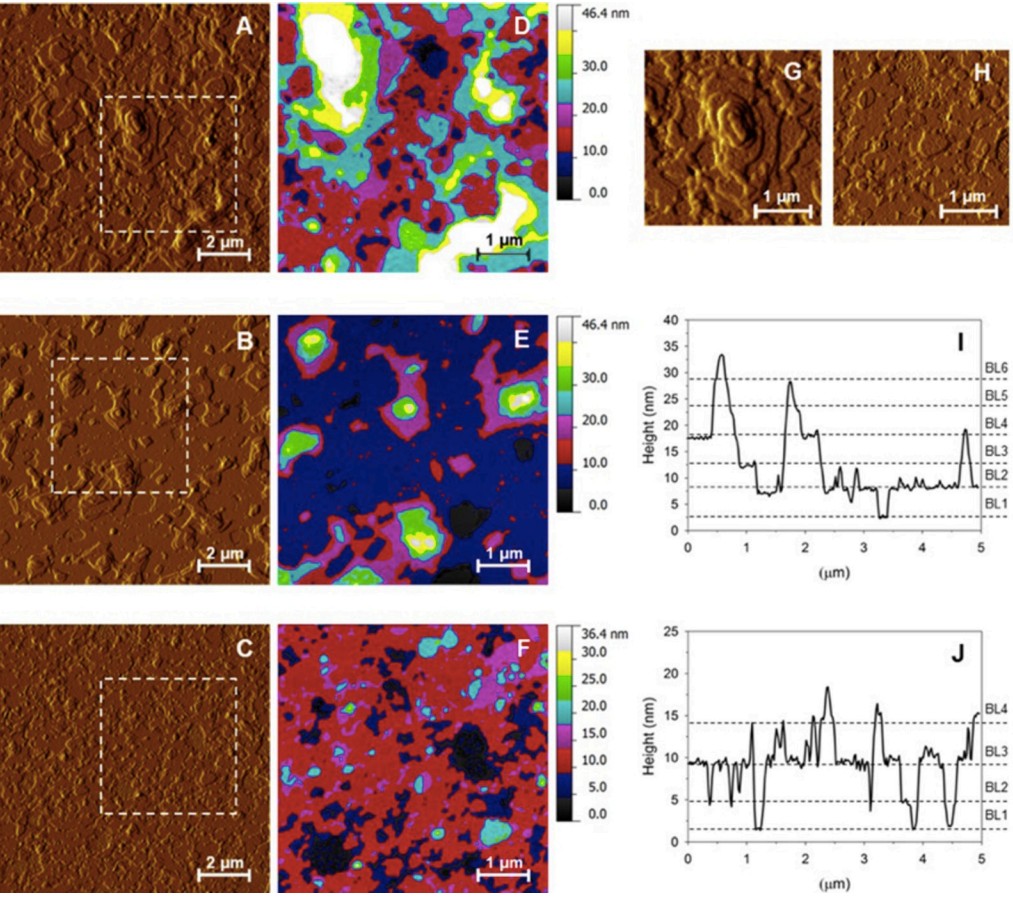

**Fig 1. Representative atomic force microscopy amplitude images of *Poractant alfa* and *Calsurf*.** Comparison of representative atomic force microscopy amplitude images of *Poractant alfa* at a phospholipid concentration of 80 mg/mL (A) and 35 mg/mL (B), and *Calsurf* at a concentration of 35 mg/mL (C). Selected regions of interest in A-C (squares delineated by white dashed lines) were color-coded in order to emphasize the height of overlapping phospholipid domains in D-F. The color switch was set every 5.5 nm for *Poractant alfa* (D-E) and 4.5 nm for *Calsurf* (F), based on the bilayer thickness measured for each surfactant preparation. In *Calsurf*, three overlapped homogeneous bilayers composed the bottom of the deposition and poor tridimensional complexity was detected. At a comparable phospholipid concentration, *Poractant alfa* showed peculiar multilamellar structures widespread above a multilayered ultrastructure. Dimensions of multilamellar structures increased with *Poractant alfa* concentration (maximum height greater than 100 nm). Detail of a lamellar body-like structure of *Poractant alfa* composed of 10 overlapped bilayers (G) and *Calsurf* ultrastructure (H). Dashed lines in (I) and (J) are height profiles marking each bilayer-bilayer (BL) transition in overlapped surfactant structure in *Poractant alfa* and *Calsurf*.

Differences in three-dimensional complexity between two surfactants were quantified by means of a surface roughness analysis. A mean peak-to-valley roughness Rtm of 28.8 ± 6.7 nm and 16.4 ± 2.5 nm was measured in *Poractant alfa* and *Calsurf*, respectively, demonstrating that surfactant preparations differed significantly in terms of microstructure ($P<0.05$).

## Langmuir Blodgett analysis

Isotherms for *Poractant alfa* and *Calsurf* were generated at 25°C, compressing symmetrically the film spread on a pure water subphase. At the starting surface pressure of 10 mN/m, the molecular areas of *Poractant alfa* and *Calsurf* were 57 Å$^2$/molecule and 48 Å$^2$/molecule, respectively, suggesting for a more expanded monolayer in the case of *Poractant alfa*. The slope of the curves in the range 10–45 mN/m, in which phospholipid chains with a low

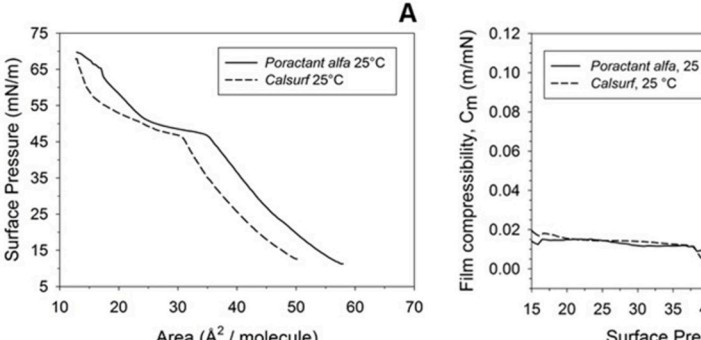

**Fig 2. *Poractant alfa* and *Calsurf* in compression isotherms and film compressibility.** (A) shows the compression isotherms of *Poractant alfa* and *Calsurf*, recorded at 25˚C using MilliQ ultra-pure water as subphase. Surfactants were spread to an initial surface pressure of ≈ 10 mN/m and subjected to symmetric compression at a rate of 40 cm$^2$/min until film collapse. (B) shows film compressibility (C$_m$) of *Poractant alfa* and *Calsurf* as a function of surface pressure. Each C$_m$ data set was calculated using compression isotherms shown in (A). C$_m$ = -1/A$^*$($\delta$A/$\delta\pi$), where A is the surface area and $\pi$ is the surface pressure.

(liquid-expanded phase) and high packing (liquid-condensed phase) coexist were equivalent for both preparations (Fig 2A). The coexistence of two phases makes surfactant films poorly compressible; indeed, the minimum value of compressibility of 0.02 m/mN was reached by both surfactants in this pressure range (Fig 2B).

At surface pressure of 45 mN/m *Poractant alfa* and *Calsurf* generated a plateau, a region of the isotherm representing the transition from monolayer to multi-layer with selective exclusion (squeeze-out) of the more fluid-like unsaturated lipids, cholesterol and proteins from a monolayer progressively enriched in DPPC. The two surfactants differed in the length of the plateau, more extended in *Calsurf*. In this region of the isotherm, surfactant films are highly compressible, due to the squeeze out of fluid molecules from the interface. In the case of *Poractant alfa* a maximum value of compressibility of 0.11 m/mN was measured, slightly higher than 0.08 m/mN in *Calsurf*, although the latter maintained a status of high compressibility for a more extended surface pressure range (from 45 to 60 mN/m). The comparison of the compressed area at a surface pressure of 68mN/m did not reach statistical significance, being 14.6 ± 0.35 Å$^2$/molecule for *Poractant alfa* and 13.8 ± 0.76 Å$^2$/molecule *for Calsurf*.

## Standardized PIP ventilation loop experiments

There were no statistical differences in body weight between the experimental groups (see S1 Table). Untreated animals (Ctrl) showed extremely low C$_{dyn}$ values (less than 0.1 mL/kg/cmH$_2$O), indicative of severe lung structural immaturity with surfactant deficiency mimicking human RDS (Fig 3). Compared to untreated animals, all surfactant-treated groups had a significantly higher C$_{dyn}$ during the first 15 min and the last 5 min of the experimental period, when PIP was 25 cmH$_2$O ($P<0.01$). Lowering PIP from 25 to 20 cmH$_2$O and then to 15 cmH$_2$O was associated with a decrease of C$_{dyn}$ in all surfactant-treated groups. At PIP levels of 20 and 15 cmH$_2$O, there were no significant differences between surfactant-treated groups and the control group, except for P200, which registered significantly higher mean C$_{dyn}$ values compared to untreated control animals ($p<0.01$ at PIP 20 cmH$_2$O and $p<0.05$ at PIP 15 cmH$_2$O). Irrespective of the surfactant preparation or the administered dose, no significant differences were found if surfactant-treated groups were compared. The response to surfactant administration in terms of C$_{dyn}$ was similar at a PEEP level of 2–3 cmH$_2$O or in the absence of PEEP.

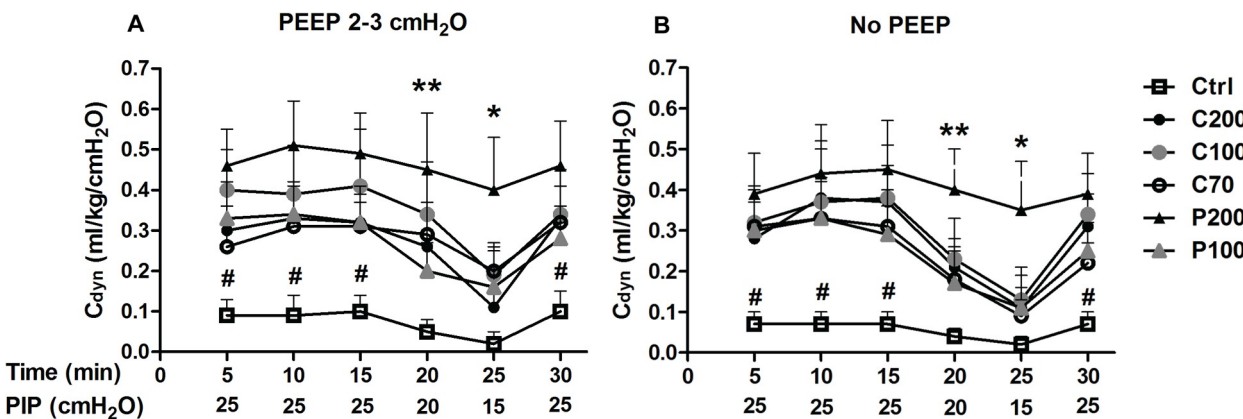

**Fig 3. *Poractant alfa* and *Calsurf* in preterm rabbits with primary surfactant deficiency in PIP loop experiment.** The dynamic compliance of the respiratory system ($C_{dyn}$) was determined during mechanical ventilation with a standardized PIP loop (25-25-25-20-15-25 $cmH_2O$, each PIP for 5 min). Two independent experimental sessions, with the PEEP set at 2–3 $cmH_2O$ (A) and without PEEP (B) were conducted. Symbols and ticks represent mean±SD. Group definitions for symbols: P200, 200 mg/kg of *Poractant alfa*, black triangles; P100, 100 mg/kg of *Poractant alfa*, grey triangles; C200, 200 mg/kg of *Calsurf*, black circles; C100, 100 mg/kg of *Calsurf*, grey circles; C70, 70 mg/kg of *Calsurf*, white circles; Ctrl, untreated animals served as controls, white squares; (n = 10 animals in each group). One-way ANOVA with S-N-K post-hoc test was used for between-group comparisons. # $P<0.01$ vs. all surfactant-treated groups; ** $P<0.01$, * $P<0.05$ vs. Ctrl group.

## Comparison of survival-time after prophylactic surfactant treatment in standardized $V_T$ ventilation experiment

A second experimental session with the same experimental groups as above, but with another 25 animals per group, was designed to investigate survival following surfactant administration. The delivery of a 200 mg/kg dose of *Poractant alfa* was associated with a significant reduction of mortality compared to untreated but identically ventilated control animals (Fig 4, $p<0.05$). After 180 min, the survival rate across groups followed the order: P200 (76%) >P100 (60%)> C200 (56%)>C100 (48%) >C70 and Ctrl (40%). However, no significant differences were detected after comparing surfactant-treated groups. The 50% survival time for the Ctrl group was around 100 min, which was extended to 130 min for the C70 group, 180 min for the C100 group and extended to over 180 min for P100, P200 and C200 groups. In our additional experiment with the same standardized $V_T$ ventilation protocol of 360 min, the 50% survival time for P200 could reach 280 min while Ctrl and C70 groups confirmed a 50% survival time around 100 and 130 min (see S1 Fig, n = 13 in each group, $P<0.05$ in log-rank (Mantel-Cox) test).

Statistically significant differences in survival rate between P200 and control groups could be already detected 45 min after the initiation of mechanical ventilation ($P<0.01$) and were maintained until the end of the experimental period (see S1 Table). Over the 180 min ventilation with standardized $V_T$, there was 50–150% increment of values of $C_{dyn}$ in all surfactant-treated groups compared to the control group (all $P<0.05$). However, no significant difference was detected between surfactant groups with different doses. At 180 min, the values of $C_{dyn}$ across groups followed the order: P200, P100>C200, C100, C70>Ctrl. (see S2 Table).

## Biochemical analysis of broncho-alveolar lavage fluid and lung tissue samples

The measurements of phospholipids and proteins in the lungs after 180-min of mechanical ventilation revealed that, there were approximately 50–60 mg/kg more of TPL, 30–35 mg/kg more of DSPC, and 80–100% relative increment of DSPC/TPL, in the lung tissue homogenates

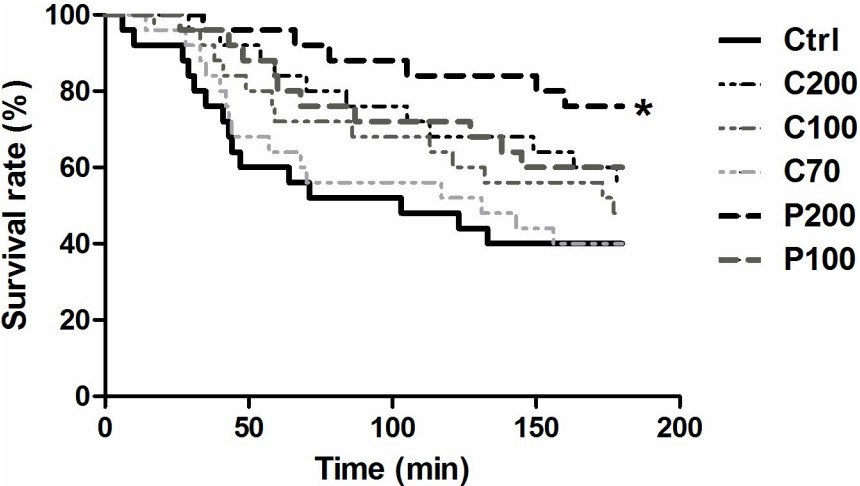

**Fig 4. Survival rate of preterm rabbits after surfactant treatment in standardized $V_T$ ventilation for 180 min.**
Lines for each group are defined as survival rate over time. At 180 min, the survival rate across groups followed the order: P200 (76%) >P100 (60%)>C200 (56%)>C100 (48%) >C70 and Ctrl (40%). The 50% survival time was 100 min in Ctrl group, 130 min in C70 group, 180 min in C100 group and more than 180 min in C200, P100 and P200 groups. P200 and P100, 200 and 100 mg/kg of *Poractant alfa*, C200, C100, and C70, 200, 100 and 70 mg/kg of *Calsurf*. Ctrl, control as untreated but ventilated. For group symbols: P200, thick black dash line; P100, thick dark grey dash line; C200, thin black dotted line; C100, thin dark grey dotted line; C70, thin light grey dotted line; Ctrl, black solid line. * $P<0.05$ vs. Ctrl in log rank (Mantel-Cox) test, n = 25 in each group.

of both P200 and C200 groups, compared with the corresponding baseline values of the control group (Fig 5A–5C). Similar trends were found for TPL and DSPC in the BAL fluid although, in this case, the DSPC/TPL ratio increased only modestly in both P200 and C200 groups (Fig 5D–5F). In contrast, values of TPL, DSPC and DSPC/TPL in P100, C100 and C70 had mild to moderate increment in both lung homogenates and BAL fluid. The amounts of TP (Fig 5G) were 30–40% higher in all mechanically ventilated animals compared to non-ventilated controls ($C_0$). The values for DSPC/TP (Fig 5H) were 2-4-fold higher in all surfactant-treated groups compared to Ctrl and $C_0$ groups. These results indicate that a higher initial phospholipid dose (e.g. 200 mg/kg) may provide a significant benefit in improving the intracellular surfactant pool.

## Lung examination

After completion of the experimental follow up, the W/D ratio was determined as an estimate of the total lung fluid capacity as well as postnatal lung fluid clearance, versus maturation. The P200 group showed the lowest W/D and was the only surfactant-treated group with significantly improved W/D values compared to untreated controls (Table 2), suggesting that lung fluid clearance was accelerated along the mechanical ventilation period in the P200 group.

$V_V$ accounts for improved alveolar expansion at the end of expiration [32] and is a good estimation of residual volume reflecting the maintenance of functional residual capacity (FRC) under various treatment. $V_V$ was the lowest in the control group, most probably due to the high incidence of atelectasis in this group. All surfactant-treated groups showed higher mean $V_V$ than controls, although only P200 and C200 reached statistical significance. CV ($V_V$) is a parameter that reflects the homogeneity of alveolar expansion. The mean CV ($V_V$) values showed a similar trend as $V_V$. Nevertheless, for CV ($V_V$), the only significant difference was observed for P200, suggesting a more homogeneous alveolar expansion after delivery of 200 mg/kg dose of *Poractant alfa*.

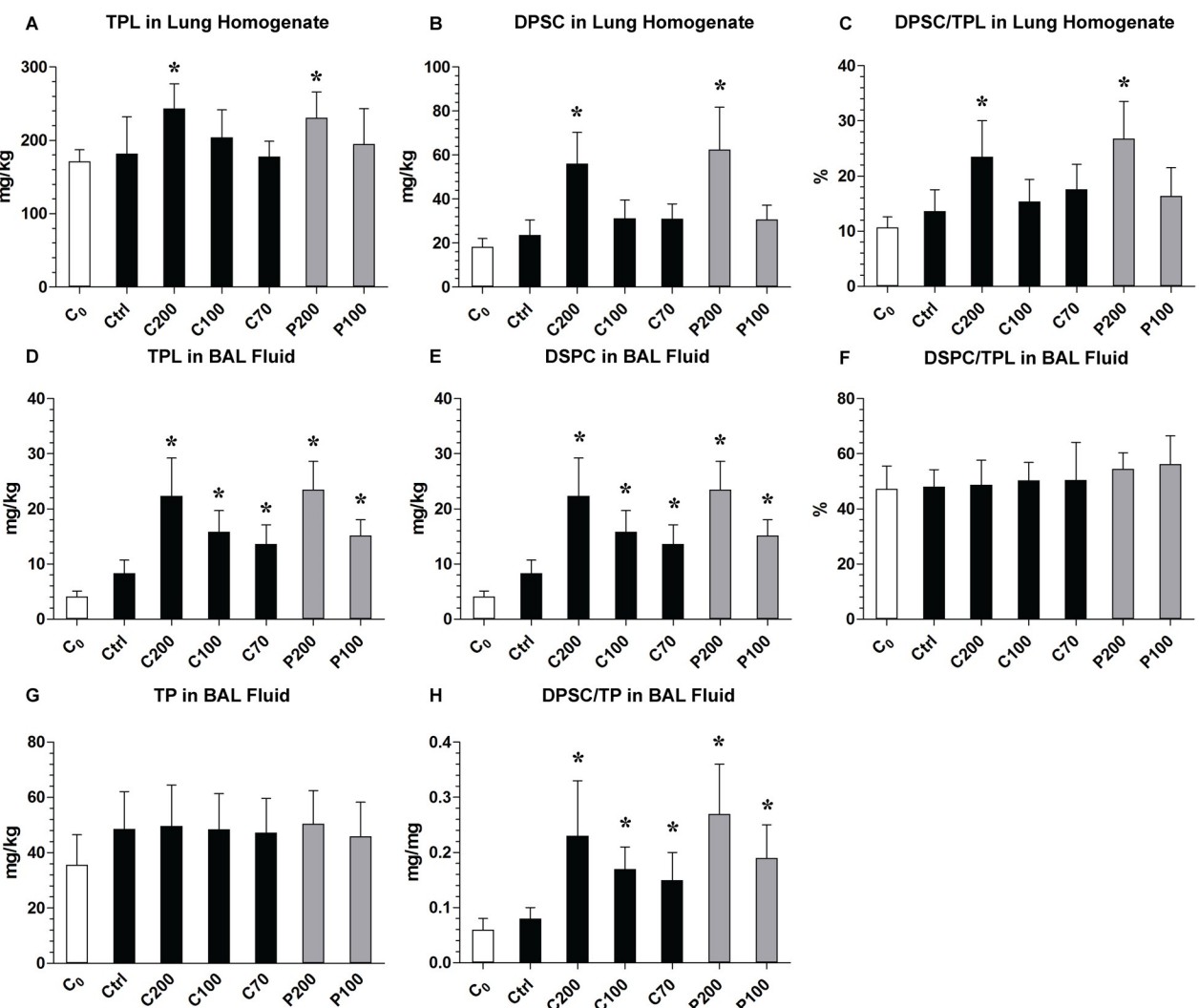

**Fig 5. Total phospholipids and proteins in lung tissue homogenates and bronchoalveolar lavage.** Comparison of total phospholipids and proteins in lung tissue homogenates (A-C) and in bronchoalveolar lavage (BAL) fluid (D-F) of preterm rabbits after treatment with *Poractant alfa* or *Calsurf* followed by 180 min standardized $V_T$ of mechanical ventilation. G-H show total proteins in BAL fluid. P200 and P100, 200 and 100 mg/kg of *Poractant alfa*, C200, C100, and C70, 200, 100 and 70 mg/kg of *Calsurf*. Ctrl, control as untreated but ventilated, and $C_0$, untreated and non-ventilated. Abbreviations: TPL, total phospholipids; DSPC, disaturated phosphatidylcholine; TP, total proteins. Values are mean±SD or ratio in percentage (%). * $P < 0.05$ vs. Ctrl and $C_0$; n = 8–10 in each group.

Lung sections were evaluated by an expert pathologist for edema, hemorrhage, neutrophil infiltration, and epithelial desquamation. All surfactant-treated groups were scored with significantly less epithelial desquamation compared to untreated control animals, which highlights the lung protective function of surfactant therapy in the setting of lung injury (Table 3). *Poractant alfa*-treated groups showed a significantly lower incidence of hemorrhage compared to controls. On the other hand, neutrophilic infiltration in both P200 and C200 was significantly lower compared to Ctrl. For comparison between two surfactants, epithelial desquamation was significantly higher in C70 than P200 group ($P < 0.05$), and P200 was significantly lower than all other four surfactant-treated groups in sum of LIS, including C200 group ($P < 0.05$ vs. C200, C100 and P100 groups; $P < 0.01$ vs. C70 group).

**Table 2. Lung conditions in preterm rabbits after 180 min of standardized $V_T$ mechanical ventilation.**

| Groups | PTX | W/D | Vv | CV [Vv] |
|---|---|---|---|---|
| P200 | 3 (12) | 6.17±1.48* | 0.50±0.06* | 0.23±0.06* |
| P100 | 6 (24) | 7.50±1.84 | 0.42±0.07 | 0.25±0.04 |
| C200 | 6 (24) | 7.78±1.57 | 0.48±0.05* | 0.27±0.04 |
| C100 | 4 (16) | 7.69±1.25 | 0.42±0.06 | 0.26±0.05 |
| C70 | 4 (16) | 8.08±1.52 | 0.38±0.06 | 0.28±0.04 |
| Ctrl | 5 (20) | 8.58±1.43 | 0.39±0.08 | 0.31±0.05 |

Group definitions: P200, *Poractant alfa* 200 mg/kg; P100, *Poractant alfa* 100 mg/kg; C200, *Calsurf* 200 mg/kg; C100, *Calsurf* 100 mg/kg; C70, *Calsurf* 70 mg/kg; Ctrl, control. Abbreviations: PTX, pneumothorax; W/D, wet-to-dry lung weight ratio; Vv, morphometry of alveolar expansion, as aerated vs. total lung parenchyma by point counting; CV [Vv], coefficient of Vv, as homogeneity of alveolar expansion. Values are number (%) for PTX, n = 25, and ratio as means±SD for the rest of variables, n = 13.

* $P<0.05$ vs. Ctrl group (one-way ANOVA with S-N-K post-hoc test).

## Discussion

In the present work, we have compared the *in vitro* and *in vivo* performance of *Poractant alfa* and *Calsurf*. *In vitro*, we found differences in the lipid composition of both preparations, which might partly explain evident ultrastructural differences as determined by AFM. *In vivo*, irrespective of the surfactant preparation and dose, we found that surfactant treatment significantly improved $C_{dyn}$, survival time, W/D and CV [Vv] compared to untreated control animals, with only P200 group reached the statistical significance. If the highest licensed clinical dose of *Poractant alfa* (200 mg/kg) and the routine clinical dose of *Calsurf* (70 mg/kg) were compared directly, treatment with *Poractant alfa* was associated with a significantly lower LIS but not in other aspects. To carry out a sound head-to-head scientific comparison between surfactants, we included additional experimental groups of *Calsurf*-treated rabbits with dosages of 100 and 200 mg/kg, which would match the licensed doses of *Poractant alfa*. Considering that *Poractant alfa* is not licensed to be administered at doses below 100 mg/kg, we did not include a group of animals treated with *Poractant alfa* at a dose of 70 mg/kg. The intratracheal delivery of either preparation at a 200 mg/kg dose was associated with a marked increase of

**Table 3. Lung injury scores of preterm rabbits after 180 min of standardized $V_T$ mechanical ventilation.**

| Groups | Lung injury scores | | | | |
|---|---|---|---|---|---|
| | Edema | Hemorrhage | Neutrophil infiltration | Epithelial desquamation | Total |
| P200 | 1.62±0.34 | 0.58±0.25* | 1.11±0.46* | 0.10±0.19** | 3.41±0.97** |
| P100 | 1.90±0.59 | 0.64±0.34* | 1.80±0.50 | 0.64±0.31** | 4.98±1.12**# |
| C200 | 1.60±0.54 | 0.98±0.34 | 1.40±0.48* | 0.50±0.33** | 4.48±1.07**# |
| C100 | 1.89±0.46 | 0.80±0.36 | 1.90±0.57 | 0.54±0.33** | 5.13±1.23**# |
| C70 | 2.10±0.56 | 1.10±0.45 | 2.10±0.36 | 0.98±0.44*# | 6.28±1.43*## |
| Ctrl | 2.40±0.46 | 1.40±0.48 | 2.50±0.48 | 2.30±0.53 | 8.60±1.96 |

Values are means±SD.

* $P<0.05$ and

** $P<0.01$ vs. Ctrl group

## $P<0.01$ and

# $P<0.05$ vs. P200 group (Kruskal-Wallis test followed by Wilcoxon-Mann-Whitney test, n = 13 in each group). P200 and P100, 200 and 100 mg/kg of *Poractant alfa*, C200, C100, and C70, 200, 100 and 70 mg/kg of *Calsurf*. Ctrl, control as untreated but ventilated.

alveolar expansion and higher intracellular DSPC pools. Nevertheless, a significant reduction of RDS-associated mortality compared to untreated but ventilated controls was only achieved after treatment with 200 mg/kg of *Poractant alfa*.

Surfactant replacement reduces RDS-associated mortality and morbidity and is widely used for the treatment of preterm infants with moderate-to-severe RDS. In clinical practice, animal-derived surfactant preparations, which contain relatively high amounts of DPPC as well as SP-B and SP-C, are still recommended over synthetic preparations.[6,7] The performance of *Poractant alfa* has been systematically compared, *in vitro* and *in vivo*, to that of other animal-derived as well as synthetic surfactants,[10,11,22,33,34] and has been confronted to other preparations in several clinical trials.[13–16] Significant differences in clinical outcomes were found between *Poractant alfa* and other bovine preparations, both from minced lung and lung lavaged fluid. However, differences were only limited to studies with a higher initial dose of *Poractant alfa*. Thus it could not be clarified whether it was related to animal source or initial dose.[35] The phospholipid concentration at which *Poractant alfa* is formulated (80 mg/mL), allowing delivering a high phospholipid dose (200 mg/kg) at relatively low fluid volumes (2.5 mL/kg), has been pointed out as an intrinsic advantage of this preparation in terms of reduction of RDS-associated mortality and re-dosing.[36] To date, no clinical studies have been conducted for comparisons with a dose of 200 mg/kg of bovine surfactants because of ethics.[18] The overall efficacy in the present study in regard to survival rate, $C_{dyn}$, biochemical measurements of intrapulmonary surfactant pool (including TPL, DSPC, DSPC/TP in BAL fluid and lung homogenate), lung morphometry and LIS was in favour of the high dose (200 mg/kg) of *Poractant alfa* not only in manufacturer-recommended doses comparison but also in amount-equivalent (i.e. mg for mg of phospholipids) comparison with *Calsurf*. We speculate that a low dose of surfactant may be still associated with its uneven distribution in the premature lungs, leading to alveolar and small airway damage and pneumothorax, and eventually to early death in the control and low-dose surfactant-treated groups. This may account for the lower LIS observed for the high dose groups compared to the low-dose groups, whereas $C_{dyn}$ was not a sensitive indicator.

We found differences in the ultrastructure as well as in the lipid composition of *Poractant alfa* and *Calsurf*. AFM analysis revealed a marked difference in the height of lipid bilayers, which were of a lower height for *Calsurf* compared to *Poractant alfa*. Such differences may be explained by the presence of cholesterol in *Calsurf*, which increases the mobility of phospholipids and decreases their packing,[37] accounting for more fluid phospholipid organization in contrast to the tightly-packed, pure phospholipid bilayers of *Poractant alfa*. Moreover, we observed multilamellar structures of up to 15 overlapping lipid bilayers in *Poractant alfa* that were absent in *Calsurf*. We speculate that such multilamellar structures in *Poractant alfa* may occur due to a higher content of DPPC and resemble the lamellar bodies observed *in vivo*. However, it cannot be ruled out that the presence/absence of such structures may also be related to different phospholipid extraction methods employed for the production of the surfactants. During tidal breathing, the structure of lamellar bodies is crucial to transfer packed lipid structures into the air-liquid interface, supporting a rapid surfactant replenishment of the respiratory interface. Highly packed lamellar bodies are less prone to inactivation by binding serum components than the more exposed single bilayers.[38] Differences in lipid composition between surfactants were also suggested by the compression isotherms. Except for the very last portion of the curve, the isotherm of *Poractant alfa* was shifted to the right compared to that of *Calsurf*. These data reflect the more expanded monolayer of *Poractant alfa* and its higher content in unsaturated phospholipids. Indeed, the presence of double bonds along the alkyl chain, not only introduces more distance between phospholipids tails, accounting for a higher surface occupied by a single phospholipid molecule, but also disturbs the monolayer

transition from a liquid expanded phase to the tightly packed liquid condensed phase (a transitional liquid-crystal state when the highest surface pressure is achieved during cyclic breathing).[39] Moreover, the more abundant fraction of unsaturated lipids in *Poractant alfa*, can support the higher value of film compressibility compared to *Calsurf*, because, upon compression, unsaturated phospholipid fraction preferentially segregates into more fluid domains easier to be compressed than those composed by saturated lipids.[40] Another difference in the compression isotherm was detectable in plateau extension, more pronounced in *Calsurf*. The plateau is representative of the removal from the interface of the more fluid surfactant components. Although unsaturated phospholipids take part to this event, the higher extension of the *Calsurf* plateau could be ascribed to the presence of cholesterol, absent in *Poractant alfa*. In fact, the plateaus differed of about 1.5-fold in our measurements, well reflecting a peculiar difference already observed upon compression among *Poractant alfa* and other commercial surfactant preparations containing cholesterol.[41]

*In vivo*, surfactant treatment was associated with significant improvement of $C_{dyn}$ and reduced lung injury, irrespective of doses and preparations. The short-term pulmonary response indicates that a phospholipid dose as low as 70 mg/kg, which is the routine dose for *Calsurf* administration, may suffice to improve lung mechanics. However, we found a significant benefit in terms of lung expansion if either *Calsurf* or *Poractant alfa* were administered at a dose of 200 mg/kg. This benefit seems to be related to the development of an intracellular surfactant pool following a high-dose surfactant administration. The amount of DSPC in BAL fluid was significantly increased in all surfactant-treated groups, which may very well explain the short-term pulmonary response observed in surfactant-treated groups compared to untreated control animals. Nevertheless, the amounts of DSPC and DSPC/TPL in lung tissue were only significantly higher in both P200 and C200 groups, suggesting that more phospholipids were taken up by alveolar epithelial cells, presumably by retaining its large aggregate form, and be readily secreted into the alveolar space, or catabolized and reutilized to synthesize and ensemble new, endogenous surfactant (containing SP-A).[42] This intracellular surfactant pool could eventually replace the "spent" alveolar surfactant, enabling a more sustained pulmonary response after a single surfactant dose, a mechanism accounting for biological half-life and re-dosing, or bioavailability, of a surfactant preparation. So far, *Calsurf* is not licensed to be delivered at a dose of 200 mg/kg, although some domestic clinical study aimed to verify its efficacy.[43] At its present phospholipid concentration (35 mg/ml), delivering 200 mg/kg of *Calsurf* to preterm infants would involve a relatively high airway fluid load, which has been associated with clinical instability.[44]

With regard to the benefits of a surfactant dose of 200 mg/kg on lung mechanism improvement, *Poractant alfa* also achieved the lowest W/D, the most uniform alveolar expansion and lowest LIS, even compared to the same dosage of *Calsurf*. We speculate that such differences between surfactants could be related to their phospholipid concentration, accounting on the volume delivered to get the same dose, and their different biochemical composition. In particular, the more abundant fraction of unsaturated phospholipids and the presence of plasmalogens in *Poractant alfa* could support surfactant absorption and transition of phospholipids from the multi-layered reservoir to the active monolayer at the air-water interface.[10,45,46] We, therefore, suspect that the significant differences observed between *Poractant alfa* administered at a dose of 200 mg/kg and untreated control animals in the PIP loop experiments, under the most astringent ventilation conditions (PIP of 15 cmH$_2$O without PEEP support), may be in part explained by the presence of plasmalogens, which may induce a fast transition from bilayer to monolayer structures under the demanding physiological breathing rates.[45] We speculate that such differences in performance between surfactants derive from differences in lipid composition and concentration, which significantly influence the ultrastructure as well

as the pulmonary performance. The results of this investigation set the rationale to look into more clinical studies which may facilitate the comparison of the two preparations based on large sample sizes.

## Conclusions

We have performed a preclinical characterization of *Calsurf* comparing its performance to that of *Poractant alfa*. Both preparations improved lung mechanics and protected lung injury in the ventilated preterm rabbit model of primary surfactant deficiency to resemble human RDS. The overall and specific effects were in a dose-dependent response pattern with maximum benefits achieved at a dose of 200 mg/kg in both preparations. *Poractant alfa* delivered at 200 mg/kg exerted better effects in terms of alveolar expansion, lung fluid clearance and protection of lung parenchyma, along with a significantly reduced mortality in comparison to untreated animals.

## Supporting information

**S1 Method. Liquid-chromatography mass-spectrometry expanded method for phospholipids analysis.**
(DOCX)

**S1 Table. Birth weight (BW) and survival rate of preterm rabbits assessed for 180 min standardized $V_T$ mechanical ventilation.** Values are mean+SD, or n (%). Group definitions: P200, *Poractant alfa* 200 mg/kg; P100, *Poractant alfa* 100 mg/kg; C200, *Calsurf* 200 mg/kg; C100, *Calsurf* 100 mg/kg; C70, *Calsurf* 70 mg/kg; Ctrl, control. ** $P<0.01$ and * $P<0.05$ vs. Ctrl group by Chi square test. Initial numbers = 25 in each group.
(DOCX)

**S2 Table. Dynamic compliance of respiratory system ($C_{dyn}$) over time in preterm rabbits with prophylactic surfactant treatment and standardized $V_T$ mechanical ventilation.** Values are mean+SD. a: *p* values are for one-way analysis of variance (ANOVA) by F test; * $p<0.05$ for between-group differences between Ctrl and any surfactant-treated groups by Student-Newmann-Keuks post hoc test. For group definitions and sample sizes at each time point see S1 Table in the Supporting information.
(DOCX)

**S1 Fig. Survival rate of preterm rabbits after prophylactic treatment in standardized $V_T$ ventilation for 360 min.** Lines for each group are defined as survival rate over time. The 50% survival time was 100, 120 and 280 min in Ctrl, C70 and P200 group, respectively. For group definition and symbols see Fig 4. * $P<0.05$ vs. Ctrl in log rank (Mantel-Cox) test, n = 13 in each group.
(TIF)

## Acknowledgments

We thank Dr. Dongmei Ding and Prof. Lian Chen for measurement and advice of lung tissue histology and morphometry.

## Author Contributions

**Conceptualization:** Bo Sun.

**Data curation:** Xiaojing Guo, Davide Amidani, Claudio Rivetti, Giuseppe Pieraccini, Silvia Catinella, Xabi Murgia, Fabrizio Salomone, Ying Dong.

**Formal analysis:** Xiaojing Guo.

**Funding acquisition:** Ying Dong.

**Investigation:** Xiaojing Guo, Siwei Luo, Davide Amidani, Giuseppe Pieraccini, Barbara Pioselli, Yaling Xu, Ying Dong.

**Methodology:** Xiaojing Guo, Siwei Luo, Davide Amidani, Barbara Pioselli, Yaling Xu.

**Project administration:** Claudio Rivetti.

**Supervision:** Ying Dong.

**Writing – original draft:** Xiaojing Guo, Davide Amidani, Barbara Pioselli, Xabi Murgia, Fabrizio Salomone.

**Writing – review & editing:** Ying Dong, Bo Sun.

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
