## [Decision Letter · Decision Letter 0]

26 Jul 2019

PONE-D-19-15621

In vitro characterization and in vivo comparison of the pulmonary outcomes of Poractant alfa (Curosurf) and Calsurf (Kelisu) in ventilated preterm rabbits

PLOS ONE

Dear Mr Sun,

Thank you for submitting your manuscript to PLOS ONE. After careful consideration, we feel that it has merit but does not fully meet PLOS ONE’s publication criteria as it currently stands. Therefore, we invite you to submit a revised version of the manuscript that addresses the points raised during the review process.

Please take into account all the remarks made by the Reviewers, especially the issue about the statistical analysis, raised by Reviewer 1. 

We would appreciate receiving your revised manuscript by Sep 09 2019 11:59PM. To enhance the reproducibility of your results, we recommend that if applicable you deposit your laboratory protocols in protocols.io, where a protocol can be assigned its own identifier (DOI) such that it can be cited independently in the future. For instructions see: http://journals.plos.org/plosone/s/submission-guidelines#loc-laboratory-protocols

We look forward to receiving your revised manuscript.

Kind regards,

Umberto Simeoni

Academic Editor

PLOS ONE

Journal Requirements:

2. At this time, we request that you  please report additional details in your Methods section regarding animal care, as per our editorial guidelines: 1) Please provide details of animal welfare for the does (e.g., shelter, food, water, environmental enrichment); 2) Please confirm that the pups were anaesthetised throughout the study and never gained consciousness; 3) Please also clarify whether the does were anaesthetised when they were euthanized and  3) Please provide the source of the porcine lungs used in this study to isolate Poractant alfa. If the pigs were euthanized specifically for this study, please include the method of euthanasia. If they were obtained from a slaughterhouse, please provide the name, location, and accreditation. Thank you for your attention to these requests.

4. Thank you for stating the following in the Financial Disclosure section:

The study was supported by Chiesi Farmaceutici S.p.A. The company contributed to the study design but had no influence in the performance, analysis, and interpretation of experimental data and in writing the manuscript. YD is a recipient of research grants

from the National Natural Science Foundation (No. 81501288) and Shanghai Municipal Commission of Health (Project Young Physician Investigator).

We note that you received funding from a commercial source:Chiesi Farmaceutici S.p.A.

Reviewers' comments:

Reviewer's Responses to Questions

**Comments to the Author**

1. Is the manuscript technically sound, and do the data support the conclusions?

Reviewer #1: Partly

Reviewer #2: Yes

Reviewer #3: Yes

Reviewer #4: Partly

2. Has the statistical analysis been performed appropriately and rigorously? 

Reviewer #1: No

Reviewer #2: Yes

Reviewer #3: Yes

Reviewer #4: No

3. Have the authors made all data underlying the findings in their manuscript fully available?

Reviewer #1: Yes

Reviewer #2: Yes

Reviewer #3: Yes

Reviewer #4: Yes

4. Is the manuscript presented in an intelligible fashion and written in standard English?

Reviewer #1: No

Reviewer #2: Yes

Reviewer #3: Yes

Reviewer #4: Yes

5. Review Comments to the Author

Reviewer #1: The tables and figures were imbedded in the manuscript instead of at the end (out of order), and the figure legends don't tell a story of what it is they are trying to show. The take home message of each figure is not clear.

The language and labeling is confusing- for example- C200 corresponds to poractant 200 mg/kg and K200 is calsurf. why would calsurf start with the K and not the C?

Also, the numbers in the tables don't always correspond with the results section. Relevance of results is not well explained.

Statistics are lacking- for example table 1.

Reviewer #2: With interest, I read the manuscript by Bo Sun et al about the characterization and comparison of two types of surfactant in preterm rabbits. The authors performed an impressive and complete vivo and vitro investigation. The study is well performed, well described and interpretable by readers with limited background in animal and in vitro studies. The study showed a favourable outcome for poractant. This study, although more detailed and done in animals, is one of many trying to identify a difference between the different surfactant products. Studies done in preterm infants have shown contradictory results, although generably in favour of poractant. Reviewing the literature, it is remarkable how much of the studies, as is this one, have been sponsored by surfactant manufacturers. It is my opinion that the surfactant research needs more independent studies. The study is a head to head experiment, eventually using 60 rabbit pups in several experiments. The study was approved by the medical ethics counsel and suffering for the animals was minimised. Thoughout the manuscript it remains somwhat unclear which rabbit underwent which investigation(s).

Some questions and suggestions remain,

How is it possible that Calsurf is marketed without any preclinical or clinical studies? For bovine or calf derived surfactant there is enough evidence and clinical studies, how does Calsurf relate to these products?

Line 351-352 PEEP or no PEEP, If I look at figure 3, I see, possibly not significant, a positive effect of PEEP on the Cdyn. The positive effect of C200 is amplified by PEEP. Would have liked to see higher PEEP levels, although not investigated in this study.

When looking furhter at figure 3, it strikes me that K100 outperforms C100 and that C100 is so much ‘worse’ than C200. Any thoughts on that?

Line 365 – 377 This seems to be an irrelevant finding and does not support the use of Poractant. The control group was not treated with surfactant. It is well know that they do not survive. Also for Table 2. Of course there is a difference between the control group and the poractant group. But this was not the object of the investigation. Also for 473 – 474 and 554-555, the control group is not under investigation.

In figure 4 it is hard to disciminate between the different lines

Reference 6 has a 2019 update

Reviewer #3: This is a bench research work comparing two surfactant preparations with a multitude of methods.Practant alfa and Calsurf are compared by various in vitro tests and also in vivo in rabbit parts. Methods include the measurement of lipid composition, atomic force microscopy, Langmuir Blodgett analysis, and in vivo effects in premature rapid pups. In the latter, compliance, survival time, and lung histology were measured. By all means, results point in the same direction, with both surfactants having typical effects, but poractant alfa function was superior. In that respect, this reviewer cannot follow the author's conclusion that large clinical study comparing both surfactants might be warranted. Such a study might be unethical in view of the superior results found in all aspects with poractant alfa, because it is likely to expose a large number of infants to an inferior surfactant preparation. Equipoise, the prerequisite of randomization, can no longer be claimed after the described results have been found.

The abstract does not really reflect what was done in the paper, and whant the resutls were. It may be misleading in referring only to Calsurf dosages of 70-100 mg per kilogram, whereas in fact, Calsurf was also tested at 200 mg per kg. Revising the abstract might be warranted.

line 477: practice, not praxis

Reviewer #4: The authors of this manuscript present a very comprehensive in vitro and in vivo analysis of these two surfactant preparations. Upon revision, these data will be an important contribution to the exogenous surfactant literature.

Major Criticisms

1. The authors use matching doses of 100 mg/kg and 200 mg/kg of both surfactants but only 70 mg/kg of Calsurf. The authors need to explain in the manuscript why there is no Poractant alfa 70 mg/kg for comparison.

2. For all experiments, the authors report statistical comparisons between experimental and control animals (which is important), but neglect to report statistical comparison between experimental groups (except for compliance (line 391) and lung injury score C200 vs K70 (line 456)). Despite this omission, they repeatedly state that one surfactant is superior to the other (lines 445, 467, 493, 553, 574, etc)

3. Methods (line 196) and associated Results/Fig 3 – There needs to be both a discussion and references to why the authors used no PEEP and a low PEEP such as 2-3 cm H2O. In part, this could be a reason why administration of surfactant did not statistically improve compliance (except for C200)(line 349, Fig 3) which is surprising. In figure 3, given the small SD for the control, it is surprising that there is no statistical difference. The authors may need to recheck their statistical analysis. For the figure 3 legend, it is not clear what “*”, “**”, and “#” refer to.

4. Methods (line 208) – For the prophylactic surfactant treatment, the authors report “A PIP ranging between 10-25 cmH2O was applied to generate a VT of 4-6 mL/kg body weight” (line 208). It is clear that pulmonary compliance changes quickly after administration of surfactant. During the 3 hour experiment, how often was the PIP titrated to maintain a goal VT of 4-6 mL/kg?

5. Figure 3 and Table S2 – There needs to be some discussion of why dynamic compliance was so different between the PIP ventilation experiment and the survival experiment. For example, at 15 minutes for C200, compliance was ~0.5 in figure 3A but was 0.28 in table S2. The authors seem to pick the data they prefer by stating in the discussion, “…was associated to significant improvement of Cdyn..” (line 531). This may be true for the data in Table S2 but not for the data in Figure 3.

6. Results – For the AFM, Langmuir Blodgett and lipid analysis, why is there no quantitative comparison or statistical analysis? How do we know if the samples are statistically different? For the lipid analysis, why were only 4 samples tested for Calsurf but 8 samples tested for Poractant alfa and why are standard deviations omitted?

7. Results (line 373) – for the survival analysis, the authors mention a 360 minute ventilation experiment but don’t include these data. These data should be included and the protocol needs to be described in the methods section.

8. Discussion – the authors should add speculation for why the lipid species are so different in Table 1, yet when given to rabbits, they are not very different in lung homogenate or BAL fluid (Figure 5).

9. Discussion – the authors should add speculation to reconcile the lack difference in compliance between surfactants yet improved survival. What is the mechanism?

10. Discussion (lines 487, 551) - The authors point out the importance of the fluid volumes of the instilled surfactant, yet don’t include this data in the methods or results. For each of the experiments presented, what volumes of surfactant were instilled?

Minor Criticisms

1. It is noted in the PLOS One competing interests statement that four of the authors are employees of Chiesi including Davide Amidani. For Davide Amidani, this needs to be noted in the manuscript itself for readers to be aware of (line 5).

2. Throughout the manuscript, the use of abbreviations based on the trade names is confusing. Consistent with the generic nomenclature, the abbreviations should be Calsurf (C70, C100, C200) and Poractant alfa (P100, P200). Trade names should be omitted from the manuscript except maybe once in the methods section.

3. For the pathology analysis (line 246), the authors need to specify whether or not the pathologist was blinded or not to the experimental group.

4. Results (line 410) – The authors state “…suggesting DSPC/TP to 410 be a sensitive indicator for lung injury protection.” Surfactants have phospholipids and surfactants can help, in some ways, to prevent lung injury. This does not mean, nor does the data provided, support that this ratio is a sensitive indicator. This statement should be omitted.

5. Results (line 437) – “VV accounts for improved alveolar expansion at the end of expiration and is a good estimate to assess the maintenance of functional residual capacity (FRC)” – a reference should be provided to support this statement.

6. Results (line 283) – DPPC is spelled wrong as “pipalmitoylphosphatidylcholine”

7. Results (line 409) – “folds” should be “fold”

8. Discussion (line 477) – While the term “praxis” may technically be correct, the term “practice” is more understood

6. PLOS authors have the option to publish the peer review history of their article (what does this mean?). If published, this will include your full peer review and any attached files.

Reviewer #1: No

Reviewer #2: No

Reviewer #3: No

Reviewer #4: No

---

## [Author Response · Author response to Decision Letter 0]

7 Sep 2019

Authors reply to each query (Q) and additional changes.

Journal Requirements:

Response: We thank the editors for the instructions. We hope that our manuscripts meet the specified requirements.

2. At this time, we request that you please report additional details in your Methods section regarding animal care, as per our editorial guidelines: 1) Please provide details of animal welfare for the does (e.g., shelter, food, water, environmental enrichment); 2) Please confirm that the pups were anaesthetised throughout the study and never gained consciousness; Please also clarify whether the does were anaesthetised when they were euthanized and 3) Please provide the source of the porcine lungs used in this study to isolate Poractant alfa. If the pigs were euthanized specifically for this study, please include the method of euthanasia. If they were obtained from a slaughterhouse, please provide the name, location, and accreditation. Thank you for your attention to these requests.

Response: 1) All does were housed until the twenty-sixth day of gestation under standard conditions of the animal center and were transported to the experiment site one day prior to the experiments with full shelter, food and water. Related contents are added to Method part, page 8, line 176-178, of R1-marked. 2) All does and pups were anaesthetized and never gained consciousness throughout the study. Does were sacrificed by overdose of potassium. Pups were euthanized by intracranial injection of 0.5 mL 2% lidocaine. Related content are add to Method part, page 8, line 187-189 and page 9, line 215-216, of R1-marked. 3) Poractant alfa was purchased by the Research and Development Department of CHIESI Farmaceutici, from the Sales Department of the same company. Therefore, all Poractant alfa vials used in this study were produced in compliance with all European regulations concerning the development of pharmaceutical products. Technically, all Poractant alfa vials are from a commercial source and no pigs were specifically euthanized for this study.

Response: S1 Fig is added to Supporting Information to show the survival rate of groups over the 360 min observation period (also see response in Q8), and the phrase has been removed in R1.

4. Thank you for stating the following in the Financial Disclosure section:

The study was supported by Chiesi Farmaceutici S.p.A. The company contributed to the study design but had no influence in the performance, analysis, and interpretation of experimental data and in writing the manuscript. YD is a recipient of research grants from the National Natural Science Foundation (No. 81501288) and Shanghai Municipal Commission of Health (Project Young Physician Investigator).

We note that you received funding from a commercial source: Chiesi Farmaceutici S.p.A.

Response: We have extended the Competing Interests Statement as follows, and is added to cover letter.

The study was supported by Chiesi Farmaceutici S.p.A, which owns the marketing rights for Poractant alfa. The company contributed to the study design but had neither influence on the performance, analysis, and interpretation of experimental data nor in writing the manuscript. BP, SC, and FS are employees of Chiesi Farmaceutici S.p.A. XM served as consultant for Chiesi in this study. YD is a recipient of research grants from the National Natural Science Foundation (No. 81501288) and Shanghai Municipal Commission of Health (Project Young Physician Investigator). This does not alter our adherence to PLOS ONE policies on sharing data and materials.

Reviewer reports: 

Reviewer #1:Q1 The tables and figures were imbedded in the manuscript instead of at the end (out of order), and the figure legends don't tell a story of what it is they are trying to show. The take home message of each figure is not clear.

Response: We thank for the reviewer’s comments. We submitted the tables and figures according to the PLOS requirements, in which the tables and figure legends were embedded in the manuscript and the figures were submitted separately as TIFF forms. The figure legends have been rephrased for your reading convenience.

Q2 The language and labeling is confusing- for example- C200 corresponds to poractant 200 mg/kg and K200 is calsurf. why would calsurf start with the K and not the C?

Response: We thank for the reviewer’s comments and feel sorry for the confusion. Reviewer 4 also mentioned this problem in Q25. Considering the generic nomenclature and to keep coincidence, the abbreviations of experimental groups are modified into C70, C100, C200 for 70, 100, 200 mg/kg Calsurf and P100, P200 for 100, 200 mg/kg Poractant alfa in R1.

Q3 Also, the numbers in the tables don't always correspond with the results section. Relevance of results is not well explained.

Response: We thank for the comments and rephrased results related to tables and figures in R1.

Q4 Statistics are lacking- for example table 1.

Response: We have analyzed and determined the surface roughness of Poractant alfa and Calsurf, which is a parameter representative of the microstructural complexity of the surfactant. We performed an ANOVA and then a t-test between Poractant alfa and Calsurf and both tests revealed a significant difference between preparations, showing a more complex three-dimensional structure of Poractant alfa.

In Table 1, we have completed the descriptive statistics by adding the standard deviation of the measurements for each class of phospholipids in the new version of the manuscript. For PL classes less represented in the surfactants, our NP HPLC-MS methods showed approximately a 20-30 % of difference in the quantitative estimation. This was in part due to the presence of some single species with intensities near the quantitative detection limit, and in part to the fact that data were acquired in full scan mode, by switching the two polarities, using an ion trap mass spectrometer. This approach, in our opinion, was adequate for the target of this work, a general comparison of the two surfactants.

Reviewer #2: With interest, I read the manuscript by Bo Sun et al about the characterization and comparison of two types of surfactant in preterm rabbits. The authors performed an impressive and complete vivo and vitro investigation. The study is well performed, well described and interpretable by readers with limited background in animal and in vitro studies. The study showed a favourable outcome for poractant. This study, although more detailed and done in animals, is one of many trying to identify a difference between the different surfactant products. Studies done in preterm infants have shown contradictory results, although generally in favour of poractant. Reviewing the literature, it is remarkable how much of the studies, as is this one, have been sponsored by surfactant manufacturers. It is my opinion that the surfactant research needs more independent studies. Q5 The study is a head to head experiment, eventually using 60 rabbit pups in several experiments. The study was approved by the medical ethics counsel and suffering for the animals was minimised. Throughout the manuscript it remains somewhat unclear which rabbit underwent which investigation(s).

Response: We feel sorry for not having clarified enough the in vivo experimental design. Actually, in total, two in vivo sessions of experiments were performed: one was standardized PIP loop ventilation experiment, and the other, standardized VT ventilation experiment.

In the PIP loop session, 60 rabbit pups were randomly allocated into six ventilation groups, namely P200 (Poractant alfa 200 mg/kg), P100 (Poractant alfa 100 mg/kg), C200 (Calsurf 200 mg/kg), C100 (Calsurf 100 mg/kg), C70 (Calsurf 70 mg/kg) and Ctrl (shame air) groups in this revised manuscript (n=10 in each group).

In the standardized VT session, in addition to the former six groups (n=25 in each group in this session), we added one non-ventilated group (C0), which served as an additional control group (n=9) for biochemical analysis of broncho-alveolar lavage and lung tissue samples.

Some questions and suggestions remain,

Q6 How is it possible that Calsurf is marketed without any preclinical or clinical studies? For bovine or calf derived surfactant there is enough evidence and clinical studies, how does Calsurf relate to these products?

Response: We appreciate this comment, which raised a relevant concern. We have conducted an additional search of articles on PubMed with the keywords “Calsurf” or “Kelisu”. As a result, we find only one additional research published: Rong Z, et al. A multicetered randomized study on early versus rescue calsurf administration for the treatment of respiratory distress syndrome in preterm infants. Am J Perinatol. 2019 Feb 4. doi:10.1055/s-0039-1678530. This publication has been added to Introduction part as reference [17]. Neither surface properties nor comparisons to other surfactants have been published yet to the best of our knowledge. We can only know Kelisu has similar formulation in terms of phospholipid concentration (roughly about 30 mg/ml) as other bovine derived surfactants (reviewed in [18] in revised manuscript). Related contents are added or rephrased in page 4, line 82-88, in R1-marked.

Q7 Line 351-352 PEEP or no PEEP, If I look at figure 3, I see, possibly not significant, a positive effect of PEEP on the Cdyn. The positive effect of C200 is amplified by PEEP. Would have liked to see higher PEEP levels, although not investigated in this study.

When looking further at figure 3, it strikes me that K100 outperforms C100 and that C100 is so much ‘worse’ than C200. Any thoughts on that?

Response: We thank for the comment. To clarify this issue, we replaced the acronyms of experimental groups in the revised manuscript (modified according to comments by Reviewer 1 and 4 in Q2 and Q25): P200 (Poractant alfa 200 mg/kg), P100 (Poractant alfa 100 mg/kg), C200 (Calsurf 200 mg/kg), C100 (Calsurf 100 mg/kg), C70 (Calsurf 70 mg/kg) and Ctrl (shame air) groups.

PEEP indeed had a positive effect on Cdyn, not only in P200 group but also in all groups including Ctrl group, with no statistic significance. However, the effect of PEEP was not the aim of this study, so we didn’t highlight this point in the manuscript. Showing both results in Fig 3A and 3B was only to clarify the similar effects of surfactants both in dose and category way. Higher PEEP was not applied because of the higher requirements of ventilation instruments.

Considering deviations, Cdyn in C100 didn’t outperform P100, and the possible reason why P100 was so much “worse” than P200 may be related to the little fluid volume of exogenous surfactant, which was only 1.25 ml/kg in P100 group (about 0.03 ml per pup assuming birth weight was 27 g). The little fluid volume may have a negative effect on surfactant diffusion in premature lung.

Q8 Line 365 – 377 This seems to be an irrelevant finding and does not support the use of Poractant. The control group was not treated with surfactant. It is well known that they do not survive. Also for Table 2. Of course there is a difference between the control group and the poractant group. But this was not the object of the investigation. Also for 473 – 474 and 554-555, the control group is not under investigation.

Response: We thank for the comments and queries. This protocol of mechanical ventilation, along with intermittent intraperitoneal injection of mixed solutions (Lidocaine+NaHCO3+Glucose), aiming at providing extra anaesthesia, energy and counterbalancing acidotic deterioration, ensured consistently the assessment of surfactant effects when the total length of ventilation may be 3-6 hours (Fig. 4 and S1 Fig. in R1). S1 Fig. is added in R1 to illustrate the survival length in the Ctrl, P200 and C70 groups. The survival length for the other three surfactant-treated groups (P100, C200 and C100) may be estimated thereupon. We applied a log-rank test as stated in the Methods (page 13, line 289) of the body text for comparison of the survival status across the groups. With this test statistic, the survival length in the surfactant-treated groups was compared step-wise in time interval with those in the control group which received identically standardized mechanical ventilation. This enabled estimation of either 50% survival (and death as reciprocal) at different time(s) or overall survival rate at a designated time over the whole ventilation period. Thus, it supports the notion that the length of survival is one of the major advantages in assessment of dose-effect of surfactant preparation.

Q9 In figure 4 it is hard to discriminate between the different lines

Response: Thanks for your comments. Lines in figure 4 have been modified as P200, Poractant alfa 200 mg/kg, thick black dash line; P100, Poractant alfa 100 mg/kg, thick dark grey dash line; C200, Calfsurf 200 mg/kg, thin black dotted line; C100, Calsurf 100 mg/kg, thin grey dotted line; C70, Calsurf 70 mg/kg, thin light grey dotted line; Ctrl, shame air, black solid line.

Q10 Reference 6 has a 2019 update

Response: We thank for the comments. Reference 6 has been updated.

Reviewer #3: This is a bench research work comparing two surfactant preparations with a multitude of methods. Practant alfa and Calsurf are compared by various in vitro tests and also in vivo in rabbit parts. Methods include the measurement of lipid composition, atomic force microscopy, Langmuir Blodgett analysis, and in vivo effects in premature rabbit pups. In the latter, compliance, survival time, and lung histology were measured. By all means, results point in the same direction, with both surfactants having typical effects, but poractant alfa function was superior.Q11 In that respect, this reviewer cannot follow the author's conclusion that large clinical study comparing both surfactants might be warranted. Such a study might be unethical in view of the superior results found in all aspects with poractant alfa, because it is likely to expose a large number of infants to an inferior surfactant preparation. Equipoise, the prerequisite of randomization, can no longer be claimed after the described results have been found.

Response: We thank the reviewer for the comments. So far, no clinical trials have been conducted to compare the effects between Poractant alfa and Calsurf in preterm infants. Besides, the effects of Poractant alfa and other bovine or calf surfactants are not clear. Meta analysis in 2017 (reference [34] in R1) concluded Poractant alfa may be better, but could not tell whether it was related to animal source or initial dose (related discussion is added to page 24, line 540-541, of R1-marked). Accordingly, page 28, line 636-639, of R1-marked, the last sentences are modified or deleted.

Q12 The abstract does not really reflect what was done in the paper, and what the results were. It may be misleading in referring only to Calsurf dosages of 70-100 mg per kilogram, whereas in fact, Calsurf was also tested at 200 mg per kg. Revising the abstract might be warranted.

Response: We thank for the comments. The arrangement and results regarding the comparison between the two surfactant preparations in abstract have been modified, as suggested.

Q13 line 477: practice, not praxis

Response: Thanks, amended.

Reviewer #4: The authors of this manuscript present a very comprehensive in vitro and in vivo analysis of these two surfactant preparations. Upon revision, these data will be an important contribution to the exogenous surfactant literature.

Major Criticisms

Q14 1. The authors use matching doses of 100 mg/kg and 200 mg/kg of both surfactants but only 70 mg/kg of Calsurf. The authors need to explain in the manuscript why there is no Poractant alfa 70 mg/kg for comparison.

Response: We thank the reviewer for the comments. From the manufacturer information, Poractant alfa is licensed to 100-200 mg/kg and Calsurf is 40-100 (average 70) mg/kg. Considering the comparisons in both licensed dose and mg-to-mg ways, we set Poractant alfa in 100 and 200 mg/kg doses, Calsurf in 70, 100 and 200 mg/kg doses, so that 100 and 200 mg/kg of two surfactants were for head-to-head comparison and Calsurf 70 mg/kg was purposely added for licensed dose comparison with Poractant alfa 100 and 200 mg/kg. Considering that Poractant alfa will never be administered at doses below 100 mg/kg, we do not see the reason to add a Poractant alfa 70 mg/kg group for comparison.

Q15 2. For all experiments, the authors report statistical comparisons between experimental and control animals (which is important), but neglect to report statistical comparison between experimental groups (except for compliance (line 391) and lung injury score C200 vs K70 (line 456)). Despite this omission, they repeatedly state that one surfactant is superior to the other (lines 445, 467, 493, 553, 574, etc)

Response: We thank for the comments and have rephrased our results in in vivo studies. In Table 3, we report significance in lung injury scores between P200 (Poractant alfa 200 mg/kg) vs. other surfactant-treated groups in the Results, page 22, line 501-507, of R1-marked. We draw the statement that Poractant alfa was superior to Calsurf for two reasons: one was the direct comparison between the two surfactants, in which LIS was significantly lower in P200 group than in all other surfactant-treated groups (Table 3); the other was the indirect comparison for the two surfactants compared to the Ctrl group, in which only P200 group could reach the statistical significance in terms of Cdyn (PIP=20 and 15 cmH2O), survival time, W/D and CV [Vv] but no shown in the other surfactant-treated groups (Fig. 3, 4 and Table 2). 

Q16 3. Methods (line 196) and associated Results/Fig 3 – There needs to be both a discussion and references to why the authors used no PEEP and a low PEEP such as 2-3 cm H2O. In part, this could be a reason why administration of surfactant did not statistically improve compliance (except for C200)(line 349, Fig 3) which is surprising. In figure 3, given the small SD for the control, it is surprising that there is no statistical difference. The authors may need to recheck their statistical analysis. For the figure 3 legend, it is not clear what “*”, “**”, and “#” refer to.

Response: We thank for the kind comments and feel sorry for the irrelevant presentation. For the Fig 3 legend, # refers to P<0.01 vs. all surfactant-treated groups; ** to P<0.01 and * to P<0.05 vs. Ctrl group. Accordingly, all surfactant-treated groups had a significantly higher Cdyn during the first 15 min and the last 5 min of the PIP loop experiment (when PIP=25 cmH2O). When lowering PIP to 20 and 15 cmH2O, only P200 group exerted significance compared to Ctrl group (P<0.01 when PIP=20 cmH2O, P<0.05 when PIP=15 cmH2O). The response to surfactant administration in terms of Cdyn was similar with or without PEEP=2-3cmH2O. The reason why the PIP loop was performed in both no- and low-PEEP was to verify whether there exists any differences of the surfactants response to PEEP. Related comments are also provided by Reviewer 1 in Q7. Two references [23, 24] and two sentences have been added in the Method part, page 9, line 217-218, and page 10, line 222, and Result part, page 17, line 379-381, and page 18, line 401-402, of R1-marked.

Q17 4. Methods (line 208) – For the prophylactic surfactant treatment, the authors report “A PIP ranging between 10-25 cmH2O was applied to generate a VT of 4-6 mL/kg body weight” (line 208). It is clear that pulmonary compliance changes quickly after administration of surfactant. During the 3 hour experiment, how often was the PIP titrated to maintain a goal VT of 4-6 mL/kg?

Response: We thank for the comments. During the first 30 min of ventilation, PIP was titrated in 3-5 minute interval, and subsequently, when the Cdyn (dynamic compliance of respiratory system) was relatively stable, PIP was adjusted at 10-15 minute interval. Related content is added to Method part of the body text, page 10, line 235-237, of R1-marked.

Q18 5. Figure 3 and Table S2 – There needs to be some discussion of why dynamic compliance was so different between the PIP ventilation experiment and the survival experiment. For example, at 15 minutes for C200, compliance was ~0.5 in figure 3A but was 0.28 in table S2. The authors seem to pick the data they prefer by stating in the discussion, “…was associated to significant improvement of Cdyn..” (line 531). This may be true for the data in Table S2 but not for the data in Figure 3.

Response: To clarify the differences of Cdyn levels (at 15 min, e.g.) in Fig 3 (as in the response to Q16) and S2 Table, different PIP was used in the PIP loop and the Vt standardized ventilation for survival. In the PIP loop ventilation, PIP=25-25-25-20-15-25 cmH2O, when PIP=25 cmH2O with or without PEEP, tidal volume could reach 8-10 ml/kg in P200 (Poractant alfa 200 mg/kg) group as a very good response while Cdyn reached ~0.5 ml/kg/cmH2O. In the survival experiment with standardized VT ventilation, PIP was adjusted in a range of 10-20 cmH2O, with PEEP at 2-3 cmH2O, to achieve 4-6 ml/kg of VT. Accordingly, Cdyn in the two different ventilation modes were not comparable. One sentence is added to Discussion, page 27, line 595, of R1-marked to clarify the issue.

Q19 6. Results – For the AFM, Langmuir Blodgett and lipid analysis, why is there no quantitative comparison or statistical analysis? How do we know if the samples are statistically different? For the lipid analysis, why were only 4 samples tested for Calsurf but 8 samples tested for Poractant alfa and why are standard deviations omitted?

Response: We have analyzed and determined the surface roughness of Poractant alfa and Calsurf, which is a parameter representative of the microstructural complexity of the surfactant. We performed an ANOVA and then a t-test comparing Poractant alfa and Calsurf. Both tests revealed a significant difference between preparations, showing a more complex three-dimensional structure of Poractant alfa. For this application, we considered the Langmuir Blodgett results as qualitative data and we did not perform any statistical analysis.

In Table 1, we have completed the descriptive statistics by adding the standard deviation of the measurements for each class of phospholipids in the new version of the manuscript. For PL classes less represented in the surfactants, our NP HPLC-MS methods showed approximately a 20-30 % of difference in the quantitative estimation. This was in part due to the presence of some single species with intensities near the quantitative detection limit, and in part to the fact that data were acquired in full scan mode, by switching the two polarities, using an ion trap mass spectrometer. This approach, in our opinion, was adequate for the target of this work, a general comparison of the two surfactants. Unfortunately, due to the limited Calsurf availability (e.g. not in Europe, only in China) we could only perform 4 LCMS runs with this particular preparation.

Q20 7. Results (line 373) – for the survival analysis, the authors mention a 360 minute ventilation experiment but don’t include these data. These data should be included and the protocol needs to be described in the methods section.

Response: We thank for comments. The additional experiment was conducted in the same standardized VT ventilation protocol except for the observation time for 360 min in P200 (Poractant alfa 200 mg/kg), C70 (Calsurf 70 mg/kg) and Ctrl (sham air) groups. We confirmed that the 50% survival time of Ctrl and C70 group was around 100 and 130 min and that of P200 group could reach 280 min. These results have been submitted as S1 Fig in Supporting information. See also on response to Q8. Related content is added to Results part, pages 18-19, lines 415-421, of R1-marked.

Q21 8. Discussion – the authors should add speculation for why the lipid species are so different in Table 1, yet when given to rabbits, they are not very different in lung homogenate or BAL fluid (Figure 5).

Response: We thank for the comments. There may exist two reasons for this speculation. For the first, although lipid species detected in Table 1 were abundant including cholesterol and several kinds of phospholipids, the percentage of saturated PL were both around 50%, which was consistent with Fig 5F though lacking significant differences among groups. For the second, as with the metabolism of exogenous surfactants in premature lungs, the surfactants were taken up by alveolar epithelial cells and reutilized as ingredients. A speculation is already presented in the Discussion, pages 25-26, lines 565-569, of R1-marked, regarding the biophysical behaviour of phospholipid composition of the two preparations. The total phospholipids (TPL) and disaturated phosphatidycholine (DSPC) detected in the experiment were those in the alveolar and intracellular pools, not identical to those from exogenous surfactant itself. In other words, these phospholipids were mixed up with endogenous phospholipids in each compartment pool.

Q22 9. Discussion – the authors should add speculation to reconcile the lack difference in compliance between surfactants yet improved survival. What is the mechanism?

Response: We thank for the comments. The facts that increment of surfactant dosage (by phospholipids) may be responsible for the prolonged survival time with association of stabilized compliance. There were not significant differences in Cdyn among surfactant-treated groups in standardized VT ventilation. We speculate that the relatively low VT should have exerted protective role in the difference of LIS in Table 3. A low dose of surfactant may be associated with its uneven distribution in the premature lungs provocative of alveolar and small airway damage and pneumothorax, leading to early death in the control and low-dose surfactant-treated groups. This is added in the Discussion, page 25, line 552-557, of R1-marked.

Q23 10. Discussion (lines 487, 551) - The authors point out the importance of the fluid volumes of the instilled surfactant, yet don’t include this data in the methods or results. For each of the experiments presented, what volumes of surfactant were instilled?

Response: We thank for the kind comments. The fluid volume for P200 (Poractant alfa 200 mg/kg), P100 (Poractant alfa 100 mg/kg), C200 (Calsurf 200 mg/kg), C100 (Calsurf 100 mg/kg) and C70 (Calsurf 70 mg/kg) groups were 2.50, 1.25, 5.71, 2.86 and 2.00 ml/kg respectively. Related contents have been added in Methods, page 9, line 205-208, of R1-marked.

Minor Criticisms

Q24 1. It is noted in the PLOS One competing interests statement that four of the authors are employees of Chiesi including Davide Amidani. For Davide Amidani, this needs to be noted in the manuscript itself for readers to be aware of (line 5).

Response: We have corrected the authors’ affiliations and we have extended the competing interest statement as:

The study was supported by Chiesi Farmaceutici S.p.A, which owns the marketing rights for Poractant alfa. The company contributed to the study design but had neither influence on the performance, analysis, and interpretation of experimental data nor in writing the manuscript. BP, SC, and FS are employees of Chiesi Farmaceutici S.p.A. XM served as consultant for Chiesi in this study. YD is a recipient of research grants from the National Natural Science Foundation (No. 81501288) and Shanghai Municipal Commission of Health (Project Young Physician Investigator). This does not alter our adherence to PLOS ONE policies on sharing data and materials.

Q25 2. Throughout the manuscript, the use of abbreviations based on the trade names is confusing. Consistent with the generic nomenclature, the abbreviations should be Calsurf (C70, C100, C200) and Poractant alfa (P100, P200). Trade names should be omitted from the manuscript except maybe once in the methods section.

Response: Thanks for the comments. The experimental groups and trade names of surfactants have been replaced accordingly in the revised manuscript.

Q26 3. For the pathology analysis (line 246), the authors need to specify whether or not the pathologist was blinded or not to the experimental group.

Response: We thank for the comments. The pathologist was blinded to the experimental design. Related content is added to Methods, page 12, line 274, of R1-marked.

Q27 4. Results (line 410) – The authors state “…suggesting DSPC/TP to be a sensitive indicator for lung injury protection.” Surfactants have phospholipids and surfactants can help, in some ways, to prevent lung injury. This does not mean, nor does the data provided, support that this ratio is a sensitive indicator. This statement should be omitted.

Response: We thank for the comments. This statement has been omitted in the revised manuscript.

Q28 5. Results (line 437) – “VV accounts for improved alveolar expansion at the end of expiration and is a good estimate to assess the maintenance of functional residual capacity (FRC)” – a reference should be provided to support this statement.

Response: Reference [31] has been provided in page 22, line 490, of R1-marked.

Q29 6. Results (line 283) – DPPC is spelled wrong as “pipalmitoylphosphatidylcholine”

Response: It has been changed.

Q30 7. Results (line 409) – “folds” should be “fold”

Response: It has been changed.

Q31 8. Discussion (line 477) – While the term “praxis” may technically be correct, the term “practice” is more understood

Response: It has been changed.

Other changes in R1 not related to reviewers’ and editors’ comments:

Minor changes about language for more readable. Reference [25-26] is added for the original method in phospholipid analysis.

---

## [Decision Letter · Decision Letter 1]

2 Dec 2019

PONE-D-19-15621R1

In vitro characterization and in vivo comparison of the pulmonary outcomes of Poractant alfa (Curosurf®) and Calsurf (Kelisu®) in ventilated preterm rabbits

PLOS ONE

Dear Mr Sun,

Thank you for submitting your revised manuscript to PLOS ONE. After consideration, we feel that minor changes or additions are still necessary before it can be published in PLoS One. 

Please address all the remaining points made by all the reviewers, with a particular attention to the additional statistical tests requested by reviewer 4, and ensure that the replies to the previous comments by reviewer 3 are complete.  

We would appreciate receiving your revised manuscript by Jan 16 2020 11:59PM. To enhance the reproducibility of your results, we recommend that if applicable you deposit your laboratory protocols in protocols.io, where a protocol can be assigned its own identifier (DOI) such that it can be cited independently in the future. For instructions see: http://journals.plos.org/plosone/s/submission-guidelines#loc-laboratory-protocols

We look forward to receiving your revised manuscript.

Kind regards,

Umberto Simeoni

Academic Editor

PLOS ONE

Reviewers' comments:

Reviewer's Responses to Questions

**Comments to the Author**

1. If the authors have adequately addressed your comments raised in a previous round of review and you feel that this manuscript is now acceptable for publication, you may indicate that here to bypass the “Comments to the Author” section, enter your conflict of interest statement in the “Confidential to Editor” section, and submit your "Accept" recommendation.

Reviewer #1: (No Response)

Reviewer #2: All comments have been addressed

Reviewer #4: (No Response)

2. Is the manuscript technically sound, and do the data support the conclusions?

Reviewer #1: Partly

Reviewer #2: Yes

Reviewer #4: Partly

3. Has the statistical analysis been performed appropriately and rigorously? 

Reviewer #1: Yes

Reviewer #2: Yes

Reviewer #4: No

4. Have the authors made all data underlying the findings in their manuscript fully available?

Reviewer #1: Yes

Reviewer #2: Yes

Reviewer #4: Yes

5. Is the manuscript presented in an intelligible fashion and written in standard English?

Reviewer #1: Yes

Reviewer #2: Yes

Reviewer #4: No

6. Review Comments to the Author

Reviewer #1: The authors have addressed many of the reviewers comments. I believe there are a few more adjustments that are necessary before publication.

1. In the body of the abstract, the authors need to include in the methods section that the company Chiesi Farmaceutici S.p.A. contributed to the study design.

2. Q14 and Q15 revision response should be added to the discussion of the manuscript.

Reviewer #2: I would like to thank the authors for their answers to my questions and wish them the best with their investigations.

Reviewer #4: The authors have addressed many, but not all, of my previous concerns. Although the original manuscript was written in acceptable English, the revision has many statements that are not intelligible and not written in standard English.

Remaining concerns:

Q15. The authors still did not perform a statistical analysis to determine if the two surfactants are different in Figures 3 and 4 and Table 2. Without this, it is not accurate to repeatedly state that one surfactant is superior to the other. An indirect comparison for the two surfactants compared to the control group is not an acceptable substitute.

Q19. The authors still did not perform a statistical analysis for the Langmuir Blodgett analysis (Figure 2) or for the Lipid analysis (Table 1). Both of these appear to very quantitative experiments. In the current manuscript, there is no way to know if there is a proven difference between surfactants for the experiments in Figure 2 and Table 1. For example, in the Langmuir Blodgett analysis, it is not valid to comment on any difference in plateau surface pressure or peak compressibility without a statistical test.

Q25. The Trade names continue to be in the article title. These should be removed.

Q28. The authors added reference 31 (Ennema JJ et al) to support the statement “VV…is a good estimate to assess the maintenance of functional residual capacity (FRC).” The referenced publication makes no mention of FRC and therefore does not support a previously proven correlation between VV and FRC.

7. PLOS authors have the option to publish the peer review history of their article (what does this mean?). If published, this will include your full peer review and any attached files.

Reviewer #1: No

Reviewer #2: Yes: Daniel Vijlbrief

Reviewer #4: No

---

## [Author Response · Author response to Decision Letter 1]

20 Dec 2019

Authors reply to the remaining query (Q) and additional changes.

Reviewer #1: The authors have addressed many of the reviewers comments. I believe there are a few more adjustments that are necessary before publication.

1. In the body of the abstract, the authors need to include in the methods section that the company Chiesi Farmaceutici S.p.A. contributed to the study design. 

Response: We take into consideration the Reviewer’s comment. According to our previous experience, in the final layout of PLOS ONE, the conflict of interest is declared in the front page of the article, just below the abstract. We have already declared our conflict of interest during the submission process, which will be accordingly displayed in the front page, provided that this work is accepted.

That said, if this would not be enough, we are willing to include the sentence suggested by Reviewer#1 in the abstract if the editor finds it more appropriate.

2. Q14 and Q15 revision response should be added to the discussion of the manuscript.

Response: Regarding Q14, we have included the following sentence in the discussion section explaining why we did not include a group of 70 mg/kg of Poractant alfa (in line 530-535, page 25, of R2-marked):

“To carry out a sound head-to-head scientific comparison between surfactants, we included additional experimental groups of Calsurf-treated rabbits with dosages of 100 and 200 mg/kg, which would match the licensed doses of Poractant alfa. Considering that Poractant alfa is not licensed to be administered at doses below 100 mg/kg, we did not include a group of animals treated with Poractant alfa at a dose of 70 mg/kg”.

Regarding Q15, we would like to clarify that, from the beggining, we always compared statistically all groups between them (including surfactant-treated groups) with the statistical methods reported in the “statistical analysis” section (line 287-288, page 13, of R2-marked). However, we only reported significant differences in the figures if P<0.05. For instance, in the ventilation loop and in the mortality curve, there were no significant differences between surfactant-treated groups; only between P200 and untreated but ventilated controls. To make this point clear for the readers, we have included new sentences in the results and discussion sections clarifying in which cases no significant differences were observed between surfactant-treated groups and also sentences indicating when the significant differences were observed between P200 and the control group:

Line 391-392, page 18, of R2-marked: “Irrespective of the surfactant preparation or the administered dose, no significant differences were found if surfactant-treated groups were compared”.

Line 418-419, page 20, of R2-marked: “However, no significant differences were detected after comparing surfactant-treated groups”.

Line 442-443, page 21, of R2-marked: “However, no significant difference was detected between surfactant groups with different doses”.

Line 537-539, page 25, of R2-marked: “Nevertheless, a significant reduction of RDS-associated mortality compared to untreated but ventilated controls was only achieved after treatment with 200 mg/kg of Poractant alfa”. 

Line 635-638, page 29, of R2-marked: “We therefore suspect that the significant differences observed between Poractant alfa administered at a dose of 200 mg/kg and untreated control animals in the PIP loop experiments…”.

Reviewer #2: I would like to thank the authors for their answers to my questions and wish them the best with their investigations.

Response: We thank the reviewer for the criticism, which has improved the quality of the manuscript.

Reviewer #4: The authors have addressed many, but not all, of my previous concerns. Although the original manuscript was written in acceptable English, the revision has many statements that are not intelligible and not written in standard English.

Response: We thank the reviewer for the advice. We have revised the manuscript and shortened and English-proofed some sentences to make the statements clearer and more comprehensible for the readers.

Remaining concerns:

Q15. The authors still did not perform a statistical analysis to determine if the two surfactants are different in Figures 3 and 4 and Table 2. Without this, it is not accurate to repeatedly state that one surfactant is superior to the other. An indirect comparison for the two surfactants compared to the control group is not an acceptable substitute.

Response: Regarding Q15, we refer to the response given to the 2nd concern of Reviewer #1. In the present version of the manuscript, we clearly state that in figures 3 and 4 and table 2 statistical analysis between all groups has been performed, and there are not significant differences between surfactant-treated groups and that the significant differences were only detected between P200 and untreated control animals. 

This has also been amended in the discussion section.

Q19. The authors still did not perform a statistical analysis for the Langmuir Blodgett analysis (Figure 2) or for the Lipid analysis (Table 1). Both of these appear to very quantitative experiments. In the current manuscript, there is no way to know if there is a proven difference between surfactants for the experiments in Figure 2 and Table 1. For example, in the Langmuir Blodgett analysis, it is not valid to comment on any difference in plateau surface pressure or peak compressibility without a statistical test.

Response: We agree with Reviewer#4 that these techniques yield quantitative data. However, we believe that the most interesting observations from Table 1 and Figure 2, and also Figure 1 (AFM) are typically qualitative (as applied already many times in the available literatures. See below additional references).

The LC-MS data revealed qualitative differences between both formulations and in particular the absence of plasmalogens in Calsurf and the absence of cholesterol in Poractant alfa. It must be noted that the analysis was performed using 1 mL of native surfactant preparations, which are formulated at concentrations of 35 mg/mL (Calsurf) and 80 mg/mL (Poractant alfa), which accounts for a 3-, 2-, 10-, and 3-fold higher PC, DPPC, PE and PI content in Poractant alfa compared to Calsurf. We believe that these data already show evident differences between both surfactants. 

Regarding the Langmuir Blodgett analysis, we have measured the value of the compressed area reached by Poractant alfa and Calsurf at a surface pressure of 68 mN/m. We have chosen 68 mN/m because both surfactants reached this very high value of pressure in these trials. 

Under these analytical conditions, the comparison between surfactants did not reach statistical significance (P>0.05), being 14.6 ± 0.35 Å2/molecule for Poractant alfa and 13.8 ± 0.76 Å2/molecule for Calsurf. We have included this comparison in the present version of the manuscript (line 376-378, page 18, of R2-marked).

That said, we believe that the shape of the isotherms is qualitative a much more relevant information and it indicates some differences between both preparations which are discussed in the manuscript. This is the reason why we included just qualitative observations in previous versions of the manuscript. 

We include below some references in which LB experiments have been used to qualitatively characterize composition and activity of pulmonary surfactants:

1. H Zhang, Q Fan, YE Wang, CR Neal, YY Zuo. Comparative study of clinical pulmonary surfactants using atomic force microscopy. Biochimica et Biophysica Acta. 2011; 1808: 1832–1842.

2. D Lukovic, A Cruz, A Gonzalez-Horta, A Almlen, T Curstedt, I Mingarro, J Pérez-Gil. Interfacial behavior of recombinant forms of human pulmonary surfactant protein SP-C Langmuir. 2012; 28: 7811−7825.

3. DM Schenck, J Fiegel. Tensiometric and phase domain behavior of lung surfactant on mucus-like viscoelastic hydrogels ACS. Appl. Mater. Interfaces. 2016; 8: 5917−5928.

4. RP Valle, CL Huang, JSC Loo, YY Zuo. Increasing hydrophobicity of nanoparticles intensifies lung surfactant film inhibition and particle retention ACS. Sustainable Chem. Eng., 2014; 2: 1574−1580.

Q25. The Trade names continue to be in the article title. These should be removed.

Response: We understand the concern of the reviewer. Following the Reviewer’s advice, we have removed the trade names of the products from the title.

Q28. The authors added reference 31 (Ennema JJ et al) to support the statement “Vv…is a good estimate to assess the maintenance of functional residual capacity (FRC).” The referenced publication makes no mention of FRC and therefore does not support a previously proven correlation between Vv and FRC.

Response: What we meant in this sentence is that Vv accounts for improved alveolar expansion at the end of expiration, as well FRC is essential to prevent alveoli collapse at the end of expiration. The method of Vv in this animal model was developed in late 1970’s and 1980’s by Dr. Bengt Robertson and his colleagues adopting Weibel’s theorem of stereometric measurement of lung volume to estimate expansion magnitude of alveoli (a gas volume approximate to FRC, which was further verified by nitrogen wash-out methods in clinical settings for human neonates in 1990’s). Different from the previous experience, the animals in our study were succeeded to be ventilated with 2-3 cmH2O PEEP in contrast to 5-6 cmH2O in clinical settings for RDS, which should have achieved a better, though still lower, level of FRC. Furthermore, the method of perfusion fixation of the animal lungs in this study may not ensure the expanded alveoli to be commensurable to FRC. Therefore, Vv should at best approximate to residual volume (RV), a part of FRC, in these premature lungs, in the presence or absence of exogenous surfactants.

Considering these, we additionally think that this parameter can be a good estimate to assess the manteinance of FRC. To clarify this concern, we have modified the position of reference 31 as follows: (Line 492-494, page 23, of R2-marked)

“VV accounts for improved alveolar expansion at the end of expiration[31] and is a good estimation of residual volume reflecting the maintenance of functional residual capacity (FRC) under various treatment”.

Additional changes in R2:

We included two short sentences and reference [20] in the methods section of the Langmuir Blodgett measurements of the main body text (Line 159-163, page 8, of R2-marked) to better understand the relationship of surface tension and surface pressure.

---

## [Editor Report · Decision Letter 2]

26 Feb 2020

In vitro characterization and in vivo comparison of the pulmonary outcomes of Poractant alfa and Calsurf in ventilated preterm rabbits

PONE-D-19-15621R2

Dear Dr. Sun,

We are pleased to inform you that your manuscript has been judged scientifically suitable for publication and will be formally accepted for publication once it complies with all outstanding technical requirements.

We ask you one last modification, i.e. to cancel the commercial names of the products from the list of the keywords.

With kind regards,

Umberto Simeoni

Academic Editor

PLOS ONE
---

## [Editor Report · Acceptance letter]

2 Mar 2020

PONE-D-19-15621R2 

*In vitro* characterization and *in vivo* comparison of the pulmonary outcomes of *Poractant alfa* and *Calsurf* in ventilated preterm rabbits 

Dear Dr. Sun:

I am pleased to inform you that your manuscript has been deemed suitable for publication in PLOS ONE. Congratulations! Your manuscript is now with our production department. 

With kind regards,

on behalf of

Dr. Umberto Simeoni 

Academic Editor

PLOS ONE